# Dependency Parsing
# is More Parameter-Efficient with Normalization

**Paolo Gajo**
University of Bologna
paolo.gajo2@unibo.it

**Domenic Rosati**
Dalhousie University
Domenic.Rosati@Dal.Ca

**Hassan Sajjad**
Dalhousie University
HSajjad@dal.ca

**Alberto Barrón-Cedeño**
University of Bologna
a.barron@unibo.it

## Abstract

Dependency parsing is the task of inferring natural language structure, often approached by modeling word interactions via attention through biaffine scoring. This mechanism works like self-attention in Transformers, where scores are calculated for every pair of words in a sentence. However, unlike Transformer attention, biaffine scoring does not use normalization prior to taking the softmax of the scores. In this paper, we provide theoretical evidence and empirical results revealing that a lack of normalization necessarily results in overparameterized parser models, where the extra parameters compensate for the sharp softmax outputs produced by high variance inputs to the biaffine scoring function. We argue that biaffine scoring can be made substantially more efficient by performing score normalization. We conduct experiments on semantic and syntactic dependency parsing in multiple languages, along with latent graph inference on non-linguistic data, using various settings of a $k$-hop parser. We train $N$-layer stacked BiLSTMs and evaluate the parser's performance with and without normalizing biaffine scores. Normalizing allows us to achieve state-of-the-art performance with fewer samples and trainable parameters. Code: https://github.com/paolo-gajo/EfficientSDP

## 1 Introduction

Dependency parsing (DP) consists in classifying node labels $t_i$ (words), edges $e_{ij}$ (relations), and edge labels $r_{ij}$ (relation types) of a dependency graph [31]. A popular model for this task, introduced by Dozat and Manning [7], entails modeling word interactions as a fully connected graph via biaffine attention. Despite its simplicity, a number of models using this biaffine transformation [3, 7, 8, 13, 14] require more parameters than necessary, due to what we identify as a lack of normalization of its outputs. We hypothesize that this overparameterization is caused by the high variance of the outputs, which extra parameters help mitigate. After showing that variance can be reduced through normalization, we demonstrate that similar or better performance can be obtained for DP with fewer parameters and training samples.

In this task, we consider a graph $\mathcal{G} = (\mathcal{V}, \mathcal{E})$ comprising a set of nodes $\mathcal{V}$, connected by a set of edges $\mathcal{E}$, with each edge $e_{ij} \in \mathcal{E}$ connecting pairs of nodes $(v_i, v_j) \in \mathcal{V}$. In latent graph inference (LGI) for dependency graphs, a sentence of $|\mathcal{V}|$ words is modeled as a $|\mathcal{V}| \times |\mathcal{V}|$ graph via biaffine scoring [7]:

$$XWX^\top = X(W_Q W_K^\top)X^\top = XW_Q(XW_K)^\top = QK^\top, \quad X \in \mathbb{R}^{|\mathcal{V}| \times d}.$$

Observe that this is equivalent to the unnormalized scores used in [38]'s self-attention: $QK^\top/\sqrt{d_k}$ where $\text{Attention}(Q, K, V) = \text{Softmax}\left(QK^\top/\sqrt{d_k}\right)V$ and $d_k$ is the output size of the keys.

39th Conference on Neural Information Processing Systems (NeurIPS 2025).

The main difference resides in the fact that Transformer attention scores are scaled by $a = 1/\sqrt{d_k}$. As explained in [38], this scaling is done because high variance inputs to the softmax function will result in large values dominating the output ($e^{x \gg 1}$) and small values decaying ($e^{x \ll 1}$). The downstream effect is exploding and vanishing gradients. To see how this works, observe that the score $s$ between any query and key vector is the result of their dot product $q_i k_i^\top$. If we assume that these vectors are zero mean and unit variance, then the variance is $\text{Var}(s) = d_k$. Finally, we get our normalization factor, which ensures that each entry in the score matrix has a standard deviation of 1, since $\text{Std} = \sqrt{d_k}$. Consequently, lower input variance will result in more stable outputs.

We observe that there is no consistency in the literature on the use of this scaling term for DP. Most works do not use this scaling term [3, 7, 8, 13, 14], with the exception of [37] who use [38]'s self-attention out of the box. In this paper, we carry out a series of experiments on semantic and syntactic DP tasks, along with non-linguistic LGI tasks, using a wide range of architecture ablations. We extend the work of Bhatt et al. [3], which represents the state of the art on dependency graph parsing in NLP.

Following the state of the art, we train stacked BiLSTMs and a biaffine classifier to infer the latent dependency graph connecting the words of a sentence. We find that, when not normalizing the scores produced by the biaffine transformation, model performance drops in terms of micro-averaged $F_1$-measure and attachment score [26]. In particular, increasing the amount of layers produces a normalization effect by reducing the variance of the output scores. Using score normalization, we find that in some cases similar or better performance can be obtained by reducing the amount of trained BiLSTM parameters by as much as 85%.

Our contributions are thus three-fold. (*i*) We show the impact of normalizing the output of the biaffine scorer in relation to the architectural changes in a DP model. (*ii*) We show DP models can obtain better performance with substantially fewer trained parameters. (*iii*) We provide a new method that substantially improves scores and obtains state-of-the-art performance for semantic [21, 46] and syntactic [24, 48] dependency parsing.

The rest of the paper is structured as follows. Section 2 presents an overview of the literature concerning DP, with specific focus on how to infer the adjacency matrix of a dependency graph. Section 3 explains why layer depth can compensate for normalization. Section 4 provides an overview of the datasets, models, evaluation, and other experimental details. Section 5 illustrates how normalization mitigates overparameterization. Finally, Section 6 draws conclusions and provides avenues for future research.

## 2    Background

LGI involves predicting all, or a proper subset of, the edges between the nodes of a graph [15, 19, 39]. For example, one could carry out LGI over molecule graphs [33] to predict the bonds between atoms or the proximity of pixel clusters in an image [9].

In the context of NLP, DP is a task concerned with inferring the dependency graph of a sentence [31]. Depending on the nature of the nodes and relations of the graph, it can be described as semantic (SemDP) [3, 8, 14, 37] or syntactic (SynDP) [7, 13, 48]. In this work, we tackle both tasks, but focus more specifically on SemDP, which can be thought of as comprising two sub-tasks: named entity recognition (NER) [23] and relation extraction (RE) [3, 14]. They can be either approached in a pipeline as separate objectives [5, 47, 50, 53] or by jointly training a single model on both sub-tasks [3, 4, 14, 49]. In the context of deep learning models trained end-to-end, three main paradigms are used: encoder-based models [3, 7, 8, 13, 14, 41, 43], decoder-only Transformers, more commonly known as large language models (LLMs) [4, 44, 51], and seq2seq encoder-decoder Transformers [17, 20, 27, 49].

In this work, we focus on encoder-style models, since they currently achieve the best performance on DP tasks [3, 14, 37] and are much more parameter efficient than LLM-based solutions. These models approach entity (node) prediction analogously to NER, while edges and relations are handled via MLP projection of node-pair features [16, 28, 29, 54] or via attention-based edge and relation inference [3, 8, 13, 14, 37]. Current state-of-the-art models, exemplified by [3], are based on the biaffine dependency parser introduced by [7], which uses stacked BiLSTM layers on top of embeddings from a pre-trained language model.

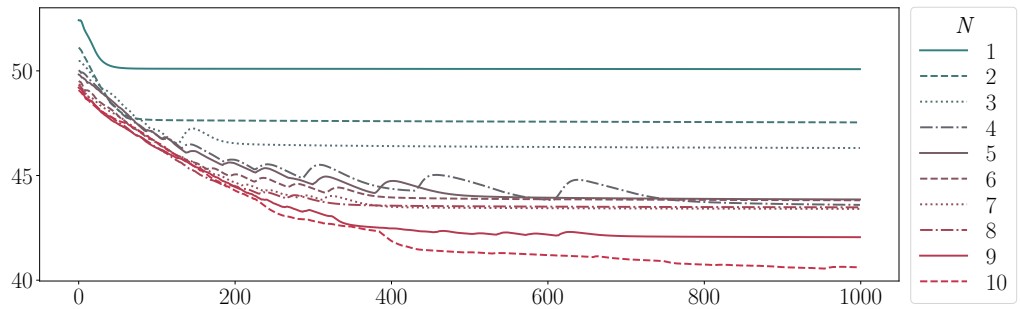

Figure 1: Effective rank $\rho(W)$ reduces over training epochs as we increase $N$ of BiLSTM layers.

## 3    Layer depth can compensate for normalization

In order to reduce the number of parameters used for biaffine scoring, we observe that the softmax function is sensitive to inputs with high variance: large values dominate the output ($e^{x \gg 1}$) and small values decay ($e^{x \ll 1}$). This causes a drop in the downstream task performance, due to some values dominating the probability outputs in the score matrix, as well as exploding and vanishing gradients. Typically, contemporary architectures based on [7] do not employ normalization, but are still able to perform well on DP tasks.

We explain this discrepancy using insights from the theory of implicit regularization [1], which contains the result that, as layer depth increases, the effective rank of the weight matrices is reduced during gradient descent. Our claim is that a reduction in the rank of the weight matrices causes a corresponding decrease in the variance of the input to the softmax function.

**Result 1** (Singular Value Dynamics Under Gradient Descent [1]). *Minimization of a loss function $\mathcal{L}(W)$ with gradient descent using weight matrices $W \in \mathbb{R}^{n \times n}$ (assuming a small learning rate $\eta$ and initialization close to the origin). An $N$-layer linear neural network leads the singular values $\sigma_r$ of $W$ to evolve in the number of iterations $t$ by:*

$$\sigma_r(t+1) \leftarrow \sigma_r(t) - \eta \cdot \langle \nabla \mathcal{L}(W(t)), \mathbf{u}_r(t)\mathbf{v}_r^\top(t) \rangle \cdot N \cdot \sigma_r(t)^{2-2/N},$$

*where $\nabla \mathcal{L}(W(t)) \in \mathbb{R}^{n \times n}$, $\mathbf{u}_r \mathbf{v}_r$ are the corresponding right and left singular vectors, and $< \cdot, \cdot >$ is the Frobenius inner product. This implies that, as the number of layers increases, the rank of the weight matrices decreases, since the smallest singular values decay.*

While Result 1 is stated for deep linear neural networks for the matrix factorization task, it has been validated empirically as applying to deep non-linear neural networks [34, 52]. We validate this finding for the biaffine scoring setting in Figure 1, which shows the inverse proportionality between the effective rank $\rho(W)$ [36] of the weights of a BiLSTM and the number $N$ of its layers.

**Claim 1** (Monotonic Increase of Output Variance with Rank). *Let $X \in \mathbb{R}^n$ be a random vector with covariance matrix $\mathbf{K}_{xx} := Cov(X) \in \mathbb{R}^{n \times n}$, and let $\mathbf{A} \in \mathbb{R}^{m \times n}$ be a fixed matrix with singular value decomposition (SVD):*

$$\mathbf{A} = \sum_{i=1}^{min(m,n)} \sigma_i \mathbf{u}_i \mathbf{v}_i^\top,$$

*where $\sigma_1 \geqslant \sigma_2 \geqslant \dots \geqslant 0$. Define the best rank-r approximation of $\mathbf{A}$ by truncated SVD as:*

$$\mathbf{A}_r := \sum_{i=1}^{r} \sigma_i \mathbf{u}_i \mathbf{v}_i^\top.$$

*Let $Y_r = \mathbf{A}_r X \in \mathbb{R}^m$ denote the image of $X$ under the rank-r linear map. Then, the total variance of $Y_r$, as measured by the trace of its covariance matrix, increases monotonically with $r$:*

$$tr(Cov(Y_r)) \leqslant tr(Cov(Y_{r+1})).$$

*Proof.* Let $\mathbf{K}_{xx} := \text{Cov}(X) \in \mathbb{R}^{n \times n}$ denote the covariance matrix of the random vector $X$. Since covariance matrices are symmetric and positive semi-definite (PSD), we have $\mathbf{K}_{xx} \geq 0$. Let $\mathbf{A}_r := \sum_{i=1}^{r} \sigma_i \mathbf{u}_i \mathbf{v}_i^\top$ be the rank-$r$ truncated SVD of $\mathbf{A}$, the covariance matrix of $Y_r := \mathbf{A}_r X$ is:

$$\text{Cov}(Y_r) = \mathbf{A}_r \mathbf{K}_{xx} \mathbf{A}_r^\top.$$

To evaluate the total variance we compute the trace:

$$\text{tr}\left(\text{Cov}(Y_r)\right) = \text{tr}\left(\mathbf{A}_r \mathbf{K}_{xx} \mathbf{A}_r^\top\right).$$

Using the cyclic property of the trace ($\text{tr}(ABC) = \text{tr}(BCA)$) and symmetry of $\mathbf{K}_{xx}$ we get:

$$\text{tr}\left(\mathbf{A}_r \mathbf{K}_{xx} \mathbf{A}_r^\top\right) = \text{tr}\left(\mathbf{K}_{xx} \mathbf{A}_r^\top \mathbf{A}_r\right).$$

Now define the matrix $\mathbf{M}_r := \mathbf{A}_r^\top \mathbf{A}_r \in \mathbb{R}^{n \times n}$, which is PSD. As $r$ increases, $\mathbf{A}_r$ includes more terms in its truncated SVD, so we have:

$$\mathbf{M}_r = \sum_{i=1}^{r} \sigma_i^2 \mathbf{v}_i \mathbf{v}_i^\top, \quad \mathbf{M}_{r+1} = \mathbf{M}_r + \sigma_{r+1}^2 \mathbf{v}_{r+1} \mathbf{v}_{r+1}^\top.$$

Thus:

$$\mathbf{M}_{r+1} \succeq \mathbf{M}_r.$$

Since $\mathbf{K}_{xx} \geq 0$, and since the trace of a product of PSD matrices respects Loewner order (i.e., if $A \preceq B$, then $\text{tr}(CA) \leq \text{tr}(CB)$ for all $C \geq 0$), we conclude:

$$\text{tr}(\mathbf{K}_{xx} \mathbf{M}_r) \leq \text{tr}(\mathbf{K}_{xx} \mathbf{M}_{r+1}),$$

which implies:

$$\text{tr}\left(\text{Cov}(Y_r)\right) \leq \text{tr}\left(\text{Cov}(Y_{r+1})\right).$$

Therefore, as the rank $r$ increases, the total variance $Y_r$ increases. $\qquad\square$

The implication of Claim 1 is that the pre-softmax score matrix from a shallow network without normalization will have a higher variance than a deeper network because of Result 1. Based on this finding, we propose to remove BiLSTM layers and use normalization —rather than having greater layer depth— in order to develop a more parameter-efficient method.

# 4 Experimental setting

## 4.1 Data

To train and evaluate our models, we use four SemDP datasets,[1] along with the 2.2 version of the Universal Dependencies English EWT treebank (enEWT) [24] and SciDTB [48] for SynDP. Table 1 summarizes datasets statistics and reports their entity and relation class annotations.

As regards SemDP, **ADE** [12] is a medical-domain dataset comprising reports of drug adverse-effect reactions. Each sample contains a single relation, "adverseEffect," which links a disease to a drug. **CoNLL04** [35] contains news texts and is annotated with the classic named entities $e_i \in \{\text{per}, \text{org}, \text{loc}\}$ and relations $r_j \in \{\text{workFor}, \text{kill}, \text{orgBasedIn}, \text{liveIn}, \text{locIn}\}$. ADE and CoNLL04 are characterized by relations which are not complex enough to form connected graphs of considerable size. **SciERC** [21] is a dataset compiled from sentences extracted from artificial intelligence literature. It is arguably the most challenging dataset used in this study, since most of the entities in the validation and testing partitions are not assigned an entity class. This makes it hard to train tag embeddings that can help to infer edges. **ERFGC** [46] is a dataset comprising "flow graphs" parsed from culinary recipes. The semantic dependency graphs of these recipes are directed and acyclic, with a single root. Differently from [3], and as advised directly by [46] (personal correspondence), we ignore the "–" edge labels present in the corpus, since they link discontinuous parts of phrasal verbs and are thus irrelevant. For ADE, CoNLL04, and SciERC we use the splits provided in [4].[2] ERFGC is not available online; we obtained it by contacting the authors of [46].

As regards SynDP, we use **enEWT** [24] to compare directly against [13]'s results. While many works evaluate SynDP on the Penn Treebank [30], it is closed-access and prohibitively expensive. Instead, we use **SciDTB** [48], a discourse analysis dataset comprising 798 abstracts extracted from the ACL Anthology.[3] It was processed for the syntax dependency parsing task using Stanza [32].

---

[1] We neglect ACE04 [22] and ACE05 [42], two popular datasets for RE, due to their prohibitive cost.

[2] `https://drive.google.com/drive/folders/1vVKJIUzK4hIipfdEGmSOCCoFmUmZwOQV`

[3] `https://aclanthology.org`

Table 1: Size of the train/dev/test splits and entity/relation classes for each dataset.

| Data | | Entities | Relations |
|---|---|---|---|
| **ADE** [12] 2,563 / 854 / 300 | | disease, drug | adverseEffect |
| **CoNLL04** [35] 922 / 231 / 288 | | organization, person, location | kill, locatedIn, workFor, orgBasedIn, liveIn |
| **SciERC** [21] 1,366 / 187 / 397 | | generic, material, method, metric, otherSciTerm, task | usedFor, featureOf, hyponymOf, evaluateFor, partOf, compare, conjunction |
| **ERFGC** [46] 242 / 29 / 29 | | food, tool, duration, quantity, actionByChef, discontAction, actionByFood, actionByTool, foodState, toolState | agent, target, indirectObject, toolComplement, foodComplement, foodEq, foodPartOf, foodSet, toolEq, toolPartOf, actionEq, timingHeadVerb, other |
| **enEWT** [24] 10,098 / 1,431 / 1,427 | | xPOS tags | UD relations |
| **SciDTB** [48] 2,567 / 814 / 817 | | xPOS tags | UD relations |

From these two datasets, we consider the xPOS tags (e.g., noun, preposition, determiner) and the syntactic dependencies (e.g., sentence root, object, adverbial modifier) to respectively be the entities and relations of the dependency graphs to infer.

In order to show that the normalization effect is language-independent, in Appendix A we also carry out SynDP experiments using Universal Dependencies [25] datasets in six other languages. Finally, we also conduct experiments on three non-linguistic datasets: PCQM-Contact [10], CIFAR10 Superpixel [9], and QM9 [33]. PCQM-Contact and QM9 are datasets which contain graphs of molecules, while CIFAR10 Superpixel contains superpixel clusters constructed from CIFAR10 images. We use these datasets to further show that the observed effects of normalization are not limited to linguistic dependency parsing, but are a general phenomenon of carrying out inference of fully connected graphs. We report the results of this additional experiment in Appendix B.

## 4.2 Model

We adopt the architecture of [3] in our work, schematized in Figure 2. It can be subdivided into four main components: encoder, tagger, parser, and decoder. The input is tokenized and passed through a BERT-like encoder, where token representations are averaged into $|\mathcal{V}|$ word-level features $\mathbf{x}_i \in \mathbb{R}^{d_f}$.[4] Optionally, additional features can be obtained by predicting the entity classes of each word with a tagger, composed by a single-layer BiLSTM $\phi$, followed by a classifier:

$$\mathbf{h}_i^{tag} = \phi(\mathbf{x}_i), \quad \mathbf{h}_i^{tag} \in \mathbb{R}^{d_h},$$
$$\mathbf{y}_i^{tag} = \text{Softmax}(\text{MLP}^{tag}(\mathbf{h}_i^{tag})), \quad \mathbf{y}_i^{tag} \in \mathbb{R}^{|T|},$$

where $T$ is the set of word tag classes. The tagger's predictions are then converted into one-hot vectors and projected into dense representations by another MLP, such that $\mathbf{e}_i^{tag} = \text{MLP}^{emb}(\mathbf{1}_T(\mathbf{y}_i^{tag}))$. These new tag embeddings are concatenated with the original BERT output and sent to the parser.

In the parser, an optional $N$-layered BiLSTM $\psi$ produces new representations $\mathbf{h}_i = \psi(\mathbf{e}_i^{tag} \oplus \mathbf{x}_i)$, which are then projected into four different representations:

$$\mathbf{e}_i^h = \text{MLP}^{(edge-head)}(\mathbf{h}_i), \quad \mathbf{e}_i^d = \text{MLP}^{(edge-dept)}(\mathbf{h}_i),$$
$$\mathbf{r}_i^h = \text{MLP}^{(rel-dept)}(\mathbf{h}_i), \quad \mathbf{r}_i^d = \text{MLP}^{(rel-head)}(\mathbf{h}_i).$$

The edge scores $s_i^{edge}$ and relation scores $s_i^{rel}$ are then calculated with the biaffine function $f$:

$$f(\mathbf{x}_1, \mathbf{x}_2; W) = \mathbf{x}_1^\top W \mathbf{x}_2 + \mathbf{x}_1^\top \mathbf{b},$$

---

[4]Using token-level representations resulted in much lower performance in preliminary experiments.

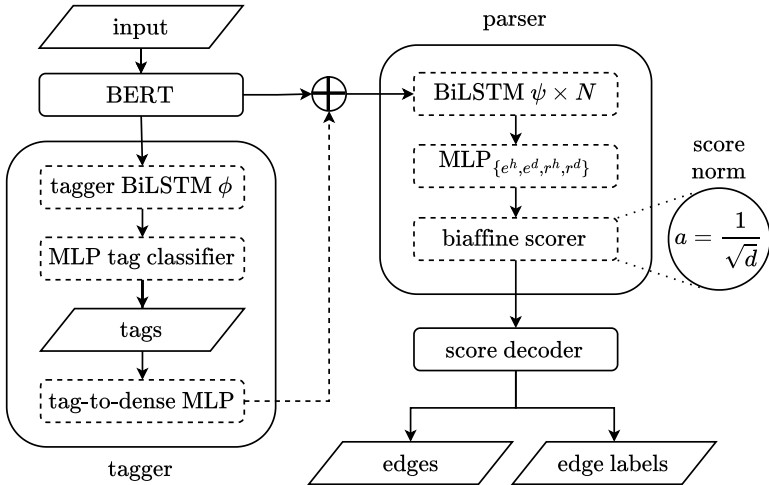

Figure 2: Dependency parsing diagram. Dashed components are the targets of ablation experiments.

$$s_i^{edge} = f^{(edge)}(\mathbf{e}_i^h, \mathbf{e}_i^d; W_e), \quad W_e \in \mathbb{R}^{d \times 1 \times d},$$
$$s_i^{rel} = f^{(rel)}(\mathbf{r}_i^h, \mathbf{r}_i^d; W_r), \quad W_r \in \mathbb{R}^{d \times |R| \times d},$$

where $R$ is the set of relation classes, i.e., the possible labels applied to an edge. In Appendix C, following [13], we extend the architecture and instead feed the representations produced by the BERT encoder into $L_\gamma \in \{0, 1, 2, 3\}$ pairs of biaffine and graph attention network (GAT) [40] layers. This allows us to see the influence of normalization when using a $k$-hop parser.

Finally, in the decoder, the edge scores are used in conjunction with the relation representations $\mathbf{r}_i^h$ and $\mathbf{r}_i^d$ to obtain the final predictions. During training, we do greedy decoding, while during inference, we use Chu-Liu/Edmonds' maximum spanning tree (MST) algorithm [11] to ensure the predictions are well-formed trees. This is especially useful with big dependency graphs, since greedy decoding is more likely to produce invalid trees as size increases. When doing greedy decoding, an edge index (i.e., an adjacency matrix) $a_i = \arg\max_j s_{ij}^{edge}$ is produced by taking the argmax of the attention scores $s_i^{edge}$ across the last dimension. The edge index is then used to select which head relation representations $\mathbf{r}_i^h$ to use to calculate the relation scores $s_i^{rel} = f(\mathbf{r}_i^h, \mathbf{r}_i^d; W), \quad W \in \mathbb{R}^{d \times |R| \times d}$. The relations are then predicted as $r_i = \arg\max_j s_{ij}^{rel}$. When using MST decoding, edge and relation scores are combined into a single energy matrix where each entry represents the score of a specific head-dependent pair with its most likely relation type. This energy matrix is then used in the MST algorithm, producing trees with a single root and no cycles. Prior to energy calculation, edge scores and relation scores are scaled so that low values are squished and high values are increased, making the log softmax produce a hard adjacency matrix.

The model is trained end-to-end jointly on the entity, edge, and relation classification objectives:

$$\mathcal{L}_{tag} = -\frac{1}{|\mathcal{V}|} \sum_{i=1}^{|\mathcal{V}|} \sum_{t=1}^{|T|} y_{i,t}^{tag} \log p\big(y_{i,t}^{tag}\big),$$

$$\mathcal{L}_{edge} = -\sum_{i,j=1}^{|\mathcal{V}|} \log p\big(y_{i,j}^{edge} = 1\big),$$

$$\mathcal{L}_{rel} = -\sum_{i=1}^{|\mathcal{V}|} \sum_{j=1}^{|\mathcal{V}|} \mathbb{1}\big(y_{i,j}^{edge} = 1\big) \sum_{\ell=1}^{|R|} y_{i,j,\ell}^{rel} \log p\big(y_{i,j,\ell}^{rel}\big),$$

$$\mathcal{L} = \lambda_1 \mathcal{L}_{tag} + \lambda_2\big(\mathcal{L}_{edge} + \mathcal{L}_{rel}\big).$$

Losses are calculated based on the gold tags, edges, and relations. We set $\lambda_1 = 0.1$ and $\lambda_2 = 1$ as hyperparameters because the tagging task is much simpler than predicting the edges, since the same top performance is always achieved regardless of any other selected architecture hyperparameters.

Table 2: Main setting hyperparameter ranges. $\psi$ = Parser BiLSTM. $f$ = biaffine layer.

| Component | Hyperparameter | Values |
|---|---|---|
| Encoder | Freeze BERT | $\nabla_{\text{BERT}} \in \{\checkmark, \times\}$ |
| Tagger | Tagger BiLSTM $\phi$ | $\phi \in \{\checkmark, \times\}$ |
| | Concat. tag embeds. | $\mathbf{e}_i^{tag} \in \{\checkmark, \times\}$ |
| Parser | $\psi$ num. layers | $N \in \{0, 1, 2, 3\}$ |
| | $\psi$ hidden dim. | $h_\psi \in \{100, 200, 300, 400\}$ |
| | MLP$^{(edge)}$ output dim. | $d_{\text{MLP}} \in \{100, 300, 500\}$ |
| | $\psi$ LayerNorm | $\text{LN}_\psi \in \{\checkmark, \times\}$ |
| | MLP$^{(edge)}$ and $f^{(edge)}$ init. | $I_{\text{par}} \sim \{\mathcal{U}, \mathcal{N}\}$ |
| | Score scaling | $a \in \{1, \frac{1}{\sqrt{d}}\}$ |

Following the usual approach for SynDP [7, 13, 14], when training on enEWT and SciDTB we use an oracle —the gold tags— and do not predict the POS tags ourselves. Since in this case we only focus on training the edge and relation classification tasks, we set $\lambda_1 = 0$.

## 4.3 Hyperparameters

We experiment with a range of hyperparameters for the encoder, tagger, and parser, as listed in Table 2. We use BERT$_{base}$ [6] as our pre-trained encoder, which we keep frozen throughout the whole training run in our main setting.

As regards the tagger, we set $L_\phi = 1$ and $h_\phi = 100$, as in [3], with weights initialized with a Xavier uniform distribution. In Appendix D.1, we ablate the $\phi$ BiLSTM and the use of the tag embeddings $\mathbf{e}_i^{tag}$ to assess their impact on overall performance. With relation to the parser, for all hyperparameter combinations of $\{N, h_\psi, d_{\text{MLP}}\}$, we run our experiments by initializing its weights with a Xavier uniform distribution ($I_{par} \sim \mathcal{U}$) and no LayerNorm $\text{LN}_\psi$. In Appendices D.2 to D.5, we conduct ablations over the parser hyperparameters indicated in Table 2.

We train our models for $2k$ steps and evaluate on the development partition of each dataset every 100 steps. For enEWT and SciDTB, we train for $5k$ steps with $1k$-step validation intervals to make our results more comparable with the state of the art [13]. We apply early stopping at 30% of the total steps without improvement and choose the best model based on the top performance on the development split. In Appendix D.6, we extend the training to $10k$ steps and fully fine-tune a variety of small and large pre-trained language models to show time-wise test performance trends more clearly. We set the learning rate at $\eta = 1 \times 10^{-3}$ when the encoder is kept frozen. In all settings, including ablations, we use AdamW [18] as the optimizer and a batch size of 8.

We use [3]'s original architecture as our baseline. It uses a frozen BERT$_{base}$ model as encoder and trains all of the components showed in Figure 2. Following the best results obtained by [7], they use three BiLSTM layers in the parser with a hidden size of 400, while the four MLPs following the stacked BiLSTM have an output size of 500 for the edge representations and 100 for the relations.

## 4.4 Evaluation

Following [3, 8, 14], we measure tagging and parsing performance on ADE, CoNLL04, SciERC, and ERFGC in terms of micro-averaged F$_1$-measure. In addition, for enEWT and SciDTB we use unlabeled (UAS) and labeled (LAS) attachment score [26]. To corroborate the validity of our results, we train and evaluate each setting, including ablations, with five random seeds. We report mean and standard deviation for the F$_1$, UAS, and LAS metrics, averaged over the five runs. To test the significance of our results, we use the one-tailed Wilcoxon signed-rank test [45].

## 5 Results and discussion

Table 3 shows the results of our experiments for the labeled edge prediction task, with the first row using the same hyperparameters as [3] ($h_\psi = 400, d_{\text{MLP}} = 500$). We also use these hyperparameters

Table 3: Micro-$F_1$ (SemDP) and LAS (SynDP) for the labeled edge prediction task. Best in bold.

| Model | $a$ | $N$ | ADE | CoNLL04 | SciERC | ERFGC | enEWT | SciDTB |
|---|---|---|---|---|---|---|---|---|
| [3] | 1 | 3 | $0.653_{\pm0.018}$ | $0.566_{\pm0.019}$ | $0.257_{\pm0.024}$ | $0.701_{\pm0.009}$ | $0.804_{\pm0.006}$ | $0.915_{\pm0.002}$ |
| Ours | 1 | 0 | $0.541_{\pm0.021}$ | $0.399_{\pm0.024}$ | $0.147_{\pm0.049}$ | $0.548_{\pm0.010}$ | $0.559_{\pm0.005}$ | $0.729_{\pm0.004}$ |
| | | 1 | $0.657_{\pm0.011}$ | $0.556_{\pm0.021}$ | $0.282_{\pm0.009}$ | $0.676_{\pm0.010}$ | $0.771_{\pm0.006}$ | $0.892_{\pm0.002}$ |
| | | 2 | $0.667_{\pm0.011}$ | $0.573_{\pm0.025}$ | $0.273_{\pm0.010}$ | $0.694_{\pm0.010}$ | $0.796_{\pm0.006}$ | $0.910_{\pm0.002}$ |
| | | 3 | $0.662_{\pm0.027}$ | $0.562_{\pm0.021}$ | $0.299_{\pm0.023}$ | $0.705_{\pm0.011}$ | $0.804_{\pm0.006}$ | $0.915_{\pm0.002}$ |
| | $\frac{1}{\sqrt{d}}$ | 0 | $0.567_{\pm0.014}$ | $0.438_{\pm0.033}$ | $0.181_{\pm0.027}$ | $0.612_{\pm0.008}$ | $0.646_{\pm0.002}$ | $0.796_{\pm0.002}$ |
| | | 1 | $0.668_{\pm0.017}$ | $0.597_{\pm0.015}$ | $0.299_{\pm0.019}$ | $0.692_{\pm0.009}$ | $0.789_{\pm0.003}$ | $0.904_{\pm0.002}$ |
| | | 2 | $0.676_{\pm0.019}$ | $0.596_{\pm0.014}$ | $0.312_{\pm0.011}$ | $0.699_{\pm0.009}$ | $0.805_{\pm0.003}$ | $0.916_{\pm0.002}$ |
| | | 3 | $\mathbf{0.686}_{\pm0.025}$ | $\mathbf{0.602}_{\pm0.017}$ | $\mathbf{0.320}_{\pm0.013}$ | $\mathbf{0.708}_{\pm0.008}$ | $\mathbf{0.807}_{\pm0.005}$ | $\mathbf{0.919}_{\pm0.001}$ |

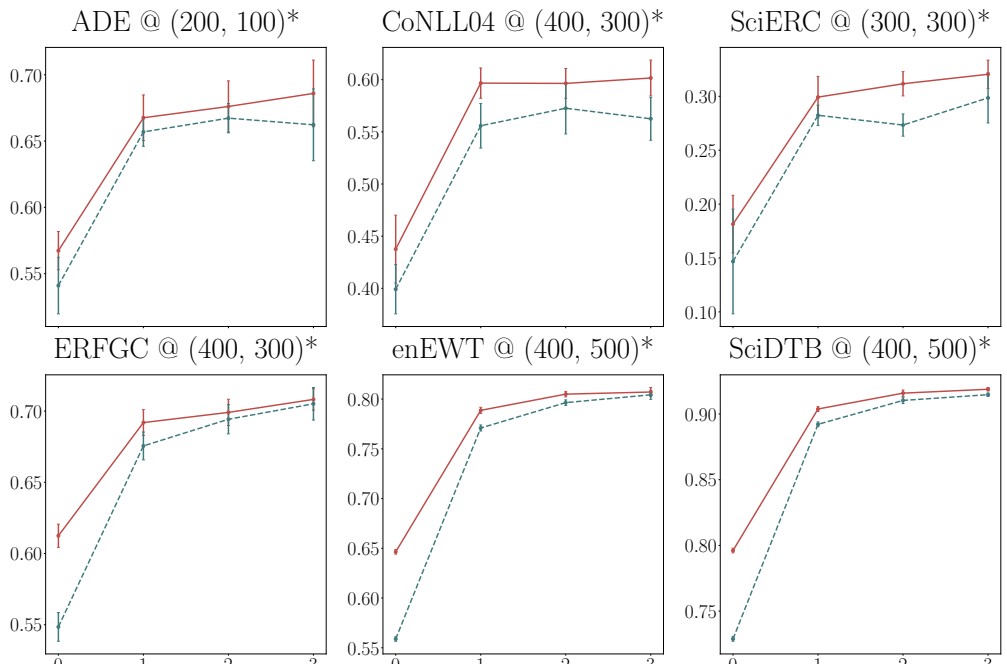

Figure 3: Micro-$F_1$ (SemDP) and LAS (SynDP) vs $N \in \{0, 1, 2, 3\}$ at $2k$ training steps. Red = norm; blue = raw. *Performance increase with normalization is statistically significant ($p < 0.01$).

for enEWT and SciDTB, since our ablation studies only concerns SemDP due to SynDP being less challenging. The lower half uses the best combinations for ADE (200, 100), CoNLL04 (400, 300), SciERC (300, 300), and ERFGC (400, 300), chosen based on mean performance across these four datasets. For brevity, the performance on the tagging and unlabeled edge prediction tasks are reported in Appendix E. Details on the used computational resources can be found in Appendix F.

Overall, ***normalizing biaffine scores provides an evident performance boost at all BiLSTM depths.*** In particular, for ERFGC we beat the state-of-the-art performance achieved by [3]. Most of the performance gain is obtained by adding the first layer, with additional ones yielding diminishing returns. In other words, the performance boost provided by score normalization is highest in the absence of the implicit normalization provided by extra parameters ($N > 0$). In general, the top performance obtained by using three BiLSTM layers and no score normalization can be matched or surpassed ***with a single BiLSTM layer, when using score normalization.*** Taking into account the lower values for $h_\psi$ and $d_{\mathrm{MLP}}$, this represents a reduction in trained parameters of up to 85%.

As laid out in Section 3, score variance tends to decay with deeper BiLSTM stacks. This in turn produces a converging trend as $N$ increases. When looking at Figure 3, this is especially evident

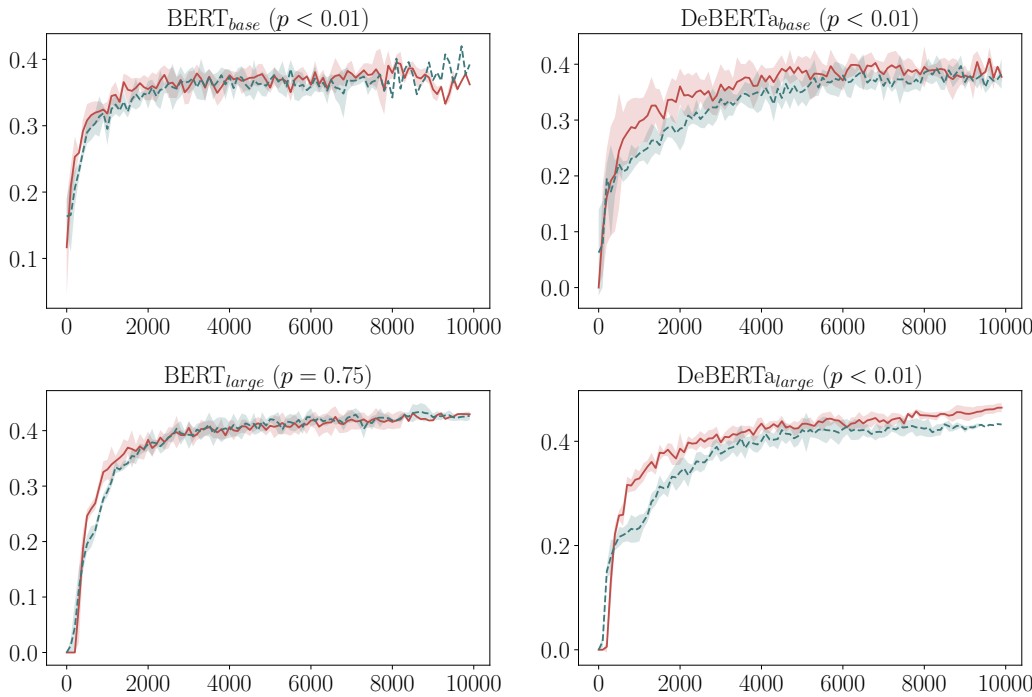

Figure 4: Test performance on SciERC (micro-averaged $F_1$-measure vs number of training steps). The $p$-values indicate greater performance with normalization (one-tailed Wilcoxon signed-rank test).

for ERFGC, enEWT, and SciDTB, for which the beneficial effect of score normalization shrinks smoothly with higher values of $N$. Although the same cannot be said for ADE, CoNLL04, and SciERC, the performance boost is still statistically significant across all layer depths ($p < 0.01$).

As regards SciERC, normalizing scores without any BiLSTM layers ($N = 0$) produces a 23% increase in performance with a strong reduction in standard deviation. SciERC arguably has the hardest dependency graphs to parse, due to the little overlap between training and testing entities, which makes it difficult to leverage tag embeddings. In addition, it is characterized by complex semantic dependencies. Therefore, we reckon normalizing biaffine scores to be especially important when dealing with hard tasks. In particular, normalizing scores helps mitigate the higher variance of the model's predictions for this challenging dataset. Conversely, the lack of score normalization exacerbates already uncertain predictions. For SciERC, this makes the trend observed in Figure 3 more unstable. Indeed, the performance first drops at $N = 2$ and then increases once again at $N = 3$, which does not happen for the other datasets.

Compared to existing approaches which do not scale the biaffine scores, fewer parameters can thus be trained to obtain similar performance. Therefore, our results indicate that parsers using raw biaffine scores are likely overparameterized, compared to the performance they could achieve by using score normalization.

In addition, score normalization increases sample efficiency by accelerating convergence. Figure 4 displays the performance on SciERC's test set for four models during full fine-tuning. As the figure shows, the speedup in convergence is statistically significant when normalizing scores. This shows how score normalization can be beneficial even when fully fine-tuning models with $\sim 10^8$ parameters, given a hard task which forces the model to make uncertain predictions.

## 6 Conclusions

In this work, we explored the effect of scaling the scores produced by biaffine transformations when predicting the edges of a dependency graph. We have demonstrated, both theoretically and empirically, that the score variance produced by a lack of score scaling hurts model performance

when predicting edges and relations. In addition, our theoretical work and experiments highlight a strong relationship between the number of trained layers and their intrinsic normalization effect.

Departing from a state-of-the-art architecture for semantic dependency parsing, we were able to improve its performance on both semantic and syntactic dependency parsing on on six datasets. On ERFGC, a dataset of directed acyclic semantic dependency graphs compiled from culinary recipes, our approach beats the state-of-the-art performance achieved by Bhatt et al. [3]. Moreover, our results showed that a single BiLSTM layer can be sufficient to match or surpass the results of state-of-the-art architectures with a decrease in trained parameters of up to 85%. In the case of SciERC, a challenging dataset for semantic dependency parsing, we found that the performance boost was particularly great when training the parser with three BiLSTM layers. Moreover, for this challenging dataset, we find that scaling the predictions of the biaffine scorer can accelerate convergence speed even when fully fine-tuning models in the $100$ to $400M$ parameter range. In addition, for three of the datasets we also observed that the performance obtained with normalized and raw scores converged smoothly as the number of trained layers increased. This supported our claim that stacking BiLSTM layers mainly serves the purpose of producing an implicit regularization, and that this effect can be obtained by normalizing the scores, without any extra parameters. In additional experiments, we corroborated our findings using data in languages other than English [25], as well as non-linguistic data, for example using a dataset of molecule graphs [33]. Using GAT layers, we further confirmed our hypothesis that GNN-based iterative adjacency refinement [13] also benefits from normalization.

In future work, we plan to extend our experiments to large-scale graph inference tasks which have been limited by the lack of parameter-efficient methods, such as long-form discourse parsing tasks. With regard to score scaling, we also wish to verify whether the positive effects of this normalization can carry over to other tasks, beyond latent graph inference. Finally, further explorations in model efficiency are warranted, given our findings, e.g., via pruning of model parameters.

## Acknowledgments

We acknowledge the support of the Killam Foundation, the Natural Sciences and Engineering Research Council of Canada (NSERC), RGPIN-2022-03943, Canada Foundation of Innovation (CFI) and Research Nova Scotia. Advanced computing resources are provided by ACENET, the regional partner in Atlantic Canada, the Digital Research Alliance of Canada, and the Vector Institute.

Paolo Gajo carried out this work while visiting Dalhousie University and working with the Hyper-matrix team. His work is funded by the EU (NextGenerationEU) with funds made available by the National Recovery and Resilience Plan (NRRP), Mission 4, Component 1, Investment 4.1 (M.D. 118/2023).

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

Table 4: Samples per train/dev/test partition for each non-English dataset.

| Dataset | Train | Dev | Test | URL |
|---|---|---|---|---|
| **Arabic** | 6,075 | 909 | 680 | `https://github.com/UniversalDependencies/UD_Arabic-PADT` |
| **Chinese** | 3,996 | 500 | 500 | `https://github.com/UniversalDependencies/UD_Chinese-GSD` |
| **Italian** | 12,161 | 538 | 467 | `https://github.com/UniversalDependencies/UD_Italian-ISDT` |
| **Japanese** | 6,824 | 493 | 518 | `https://github.com/UniversalDependencies/UD_Japanese-GSD` |
| **Spanish** | 13,821 | 1,607 | 1,666 | `https://github.com/UniversalDependencies/UD_Spanish-AnCora` |
| **Wolof** | 1,149 | 436 | 453 | `https://github.com/UniversalDependencies/UD_Wolof-WTB` |

Table 5: SynDP results for different languages, with and without score normalization.

| $a$ | $N$ | AR | CH | IT | JP | ES | WO |
|---|---|---|---|---|---|---|---|
| $1$ | 0 | $0.538_{\pm 0.005}$ | $0.395_{\pm 0.007}$ | $0.563_{\pm 0.006}$ | $0.493_{\pm 0.010}$ | $0.554_{\pm 0.004}$ | $0.252_{\pm 0.007}$ |
| | 1 | $0.723_{\pm 0.008}$ | $0.653_{\pm 0.007}$ | $0.792_{\pm 0.003}$ | $0.812_{\pm 0.003}$ | $0.775_{\pm 0.003}$ | $0.525_{\pm 0.006}$ |
| | 2 | $0.745_{\pm 0.004}$ | $0.710_{\pm 0.005}$ | $0.826_{\pm 0.003}$ | $0.844_{\pm 0.005}$ | $0.807_{\pm 0.002}$ | $0.587_{\pm 0.007}$ |
| | 3 | $0.748_{\pm 0.004}$ | $0.717_{\pm 0.007}$ | $0.832_{\pm 0.002}$ | $0.849_{\pm 0.003}$ | $0.808_{\pm 0.002}$ | $0.614_{\pm 0.005}$ |
| $\frac{1}{\sqrt{d}}$ | 0 | $0.609_{\pm 0.003}$ | $0.479_{\pm 0.007}$ | $0.633_{\pm 0.002}$ | $0.585_{\pm 0.005}$ | $0.620_{\pm 0.002}$ | $0.305_{\pm 0.005}$ |
| | 1 | $0.737_{\pm 0.007}$ | $0.693_{\pm 0.003}$ | $0.820_{\pm 0.004}$ | $0.838_{\pm 0.004}$ | $0.801_{\pm 0.003}$ | $0.556_{\pm 0.006}$ |
| | 2 | $0.758_{\pm 0.003}$ | $0.736_{\pm 0.005}$ | $0.842_{\pm 0.004}$ | $0.859_{\pm 0.004}$ | $0.822_{\pm 0.001}$ | $0.613_{\pm 0.008}$ |
| | 3 | $\mathbf{0.759}_{\pm 0.005}$ | $\mathbf{0.742}_{\pm 0.006}$ | $\mathbf{0.845}_{\pm 0.002}$ | $\mathbf{0.859}_{\pm 0.003}$ | $\mathbf{0.823}_{\pm 0.002}$ | $\mathbf{0.633}_{\pm 0.005}$ |

## A Non-English data results

In this appendix, we carry out experiments on the SynDP task on six non-English datasets, all downloaded from Universal Dependencies.[5] The datasets are listed in Table 4 and comprise the following languages: Arabic, Chinese, Italian, Japanese, Spanish, and Wolof. The results of the experiments are reported in Table 5. Since the data is in different languages, as encoder we use mBERT, the multilingual version of BERT.[6] All model hyperparameters are kept the same as in the main setting.

Similarly to the English results observed for enEWT and SciDTB, the addition of BiLSTM layers increases the performance. Furthermore, additional layers have a diminished boosting effect on the performance. This shows that the benefits of normalization are independent of the language of which the dependencies are being parsed.

## B Non-linguistic results

In this appendix we lay out the unlabeled latent graph inference experiments we carried out on three non-linguistic datasets: PCQM-Contact [10], CIFAR10 Superpixel [9], and QM9 [33]. PCQM-Contact and QM9 are molecule datasets, while CIFAR10 Superpixel is a dataset of superpixel clusters constructed from CIFAR10. Nodes in QM9 and PCQM-Contact respectively contain 9 and 11 features describing each atom in a molecule. In CIFAR10 Superpixel, nodes contain the 3D RGB features of each pixel cluster. In QM9 and CIFAR10 Superpixel, we concatenate the 3D and 2D spatial coordinates of the nodes to their features. Therefore, the resulting node feature dimensionalities of the three datasets are: $d_{PCQM} = 9$, $d_{CIFAR10} = 5$, and $d_{QM9} = 14$.

We feed the graph node features into $L_\gamma \in \{1, 2, 3\}$ GAT layers and provide them with a fully-connected adjacency matrix without any edge attributes. Although the adjacency is fully connected, meaning each node can attend to all others in a single step, stacking $L_\gamma$ layers refines the globally aggregated representations over 1–3 iterations. The processed node features are passed into the biaffine layer to produce edge predictions. In this case, we pass the scores into a sigmoid function

---

[5]`https://universaldependencies.org`
[6]`https://huggingface.co/google-bert/bert-base-multilingual-cased`

Table 6: Results for the non-linguistic datasets at varying depths of GAT layers.

| $a$ | $L_\gamma$ | PCQM-Contact | CIFAR10 Superpixel | QM9 |
|---|---|---|---|---|
| | 1 | $0.241_{\pm0.044}$ | $0.407_{\pm0.110}$ | $0.861_{\pm0.008}$ |
| 1 | 2 | $0.217_{\pm0.019}$ | $0.528_{\pm0.134}$ | $0.876_{\pm0.006}$ |
| | 3 | $0.245_{\pm0.034}$ | $\mathbf{0.751}_{\pm0.014}$ | $0.877_{\pm0.008}$ |
| | 1 | $\mathbf{0.288}_{\pm0.090}$ | $0.531_{\pm0.038}$ | $0.886_{\pm0.001}$ |
| $\frac{1}{\sqrt{d}}$ | 2 | $0.209_{\pm0.028}$ | $0.585_{\pm0.121}$ | $0.880_{\pm0.004}$ |
| | 3 | $0.237_{\pm0.019}$ | $0.703_{\pm0.046}$ | $\mathbf{0.881}_{\pm0.013}$ |

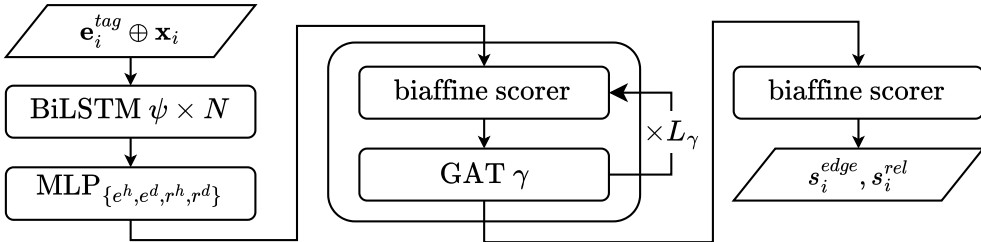

Figure 5: Extended parser architecture, featuring $L_\gamma \in \{0,1,2,3\}$ pairs of biaffine and GAT layers.

because we are not trying to predict an arborescence with the maximum spanning tree algorithm like in dependency parsing. We use $N = 0$ BiLSTM layers to isolate the effect of the GAT layers with respect to any other network components. Scores are averaged over three different seeds.

The unlabeled $F_1$ scores are reported in Table 6. In all cases, the results show that the gap in performance is large at $L_\phi = 1$ and decreases (or flips) with additional layers. This corroborates the findings from the other settings and is an especially important result considering the phenomenon holds even for such a small network, consisting of just $100$–$160k$ parameters.

## C    GAT results

In this appendix, we extend the main-setting parser, following [13], and carry out $k$-hop parsing experiments by passing the node representations through $L_\gamma \in \{0,1,2,3\}$ pairs of biaffine and GAT layers. These allow the model to encode higher-order dependencies before the final biaffine layer. Figure 5 depicts the extended architecture. We keep $N = 0$ to isolate the effect of the GAT layers.

As Table 7 shows, the results of this architecture are consistent with the ones obtained using the main-setting 1-hop parser, with the performance increasing across the board when using normalization. As regards the number of GAT layers, performance increases when using $L_\gamma \in \{1,2\}$ layers, compared to $L_\gamma = 0$, but drops again for all datasets at $L_\gamma = 3$. In general, the best performance is obtained with normalization and $L_\gamma \in \{0,1\}$ layers. It is particularly interesting to notice that for enEWT the top performance without normalization is obtained at $L_\gamma = 0$, while $L_\gamma = 1$ provides better performance when normalizing biaffine scores. Overall, the usefulness of multi-hop dependency parsing for SemDP is scarce if compared with the SynDP results, where higher-order dependencies provide more useful information for the predictions.

## D    Ablations

This appendix provides a collection of ablation experiments for various components and settings of the model used in the main experiments.

### D.1    Tagger

Table 8 reports the results for the best values of $h_\psi$ and $d_{\mathrm{MLP}}$ (chosen as described in Section 4), ablating over $\phi \in \{\checkmark, \times\}$ and $\mathbf{e}_i^{tag} \in \{\checkmark, \times\}$. The first quarter of the table is equivalent to Table 3.

Table 7: Micro-F$_1$ (SemDP) and LAS (SynDP) for the labeled edge prediction task at varying depths of $L_\gamma \in \{0, 1, 2, 3\}$ GAT layers. Best in bold.

| $a$ | $L_\gamma$ | ADE | CoNLL04 | SciERC | ERFGC | enEWT | SciDTB |
|---|---|---|---|---|---|---|---|
| 1 | 0 | $0.517_{\pm 0.038}$ | $0.526_{\pm 0.046}$ | $0.123_{\pm 0.050}$ | $0.536_{\pm 0.013}$ | $0.610_{\pm 0.009}$ | $0.727_{\pm 0.006}$ |
| | 1 | $0.509_{\pm 0.022}$ | $0.493_{\pm 0.037}$ | $0.039_{\pm 0.020}$ | $0.599_{\pm 0.008}$ | $0.583_{\pm 0.011}$ | $0.731_{\pm 0.009}$ |
| | 2 | $0.509_{\pm 0.025}$ | $0.485_{\pm 0.033}$ | $0.029_{\pm 0.041}$ | $0.611_{\pm 0.015}$ | $0.540_{\pm 0.012}$ | $0.710_{\pm 0.013}$ |
| | 3 | $0.487_{\pm 0.040}$ | $0.481_{\pm 0.087}$ | $0.000_{\pm 0.000}$ | $0.585_{\pm 0.009}$ | $0.511_{\pm 0.015}$ | $0.700_{\pm 0.010}$ |
| $\frac{1}{\sqrt{d}}$ | 0 | $0.556_{\pm 0.037}$ | $\mathbf{0.574}_{\pm 0.028}$ | $\mathbf{0.156}_{\pm 0.031}$ | $0.607_{\pm 0.009}$ | $0.671_{\pm 0.007}$ | $0.778_{\pm 0.006}$ |
| | 1 | $\mathbf{0.587}_{\pm 0.015}$ | $0.550_{\pm 0.036}$ | $0.148_{\pm 0.028}$ | $\mathbf{0.639}_{\pm 0.007}$ | $\mathbf{0.696}_{\pm 0.005}$ | $\mathbf{0.809}_{\pm 0.008}$ |
| | 2 | $0.534_{\pm 0.023}$ | $0.563_{\pm 0.029}$ | $0.128_{\pm 0.029}$ | $0.637_{\pm 0.007}$ | $0.657_{\pm 0.007}$ | $0.795_{\pm 0.008}$ |
| | 3 | $0.543_{\pm 0.024}$ | $0.514_{\pm 0.026}$ | $0.071_{\pm 0.021}$ | $0.616_{\pm 0.006}$ | $0.596_{\pm 0.010}$ | $0.770_{\pm 0.007}$ |

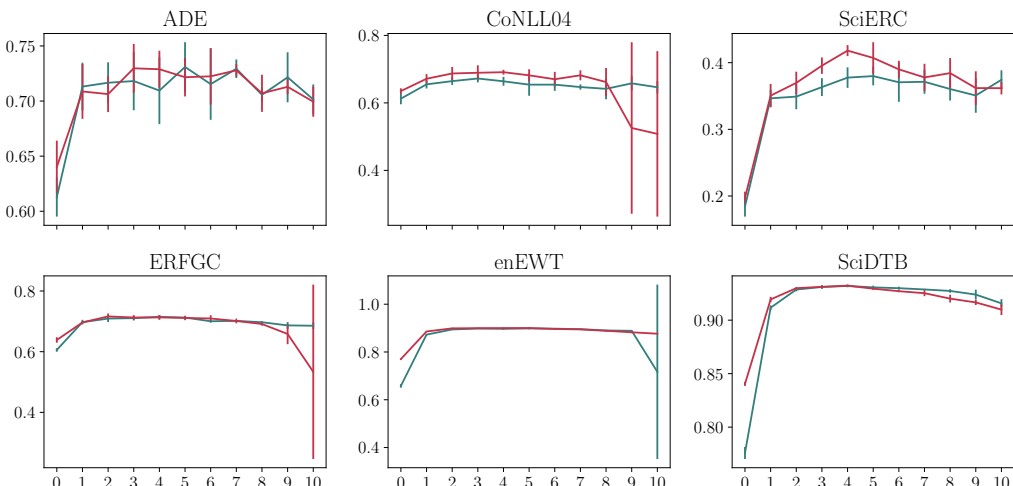

Figure 6: Micro-F$_1$ (SemDP) and LAS (SynDP) vs $N \in \{0, \dots, 10\}$ at $20k$ training steps. Best per-dataset hyperparameters ($h_\psi$, $d_{\mathrm{MLP}}$) with $\phi = \checkmark$ and $\mathbf{e}_i^{tag} = \checkmark$. Red = norm; blue = raw.

Using both the tagger BiLSTM and tag embeddings on average produces the best results. This is sensible, since a better tagger produces better tag embeddings, which in turn help inform the edge and relation classification tasks. The model in this case obtains the best performance on ADE, CoNLL04, and ERFGC. Its top performance for SciERC (0.320) is also close to the overall peak performance (0.324), obtained without using the tagger BiLSTM. On average, the mean performance is considerably higher when combining the positive contributions of the BiLSTM and the tag embeddings.

### D.2 Additional BiLSTM layers

Figure 6 depicts performance against the number of BiLSTM layers $N$ at $20k$ training steps. Compared with the main setting's $2k$ steps, we increase the amount of training because lower amounts resulted in very high F$_1$ standard deviations with as few as $N = 6$ layers in preliminary experiments. With a shallow stack of BiLSTM layers ($N < 6$), performance is greater when normalizing scores. However, at deeper depths performance becomes very unstable for CoNLL04, ERFGC, and enEWT. Nonetheless, it is evident that score normalization is beneficial, with deeper stacks of BiLSTM layers reducing the effect of normalization, supporting our main-setting claims and experiments.

As observed in Section 5, adding trainable layers seemingly allows the parameters to scale the variance of the scores to compensate for the lack of explicit normalization. As reported in Appendix D.6, a similar behavior can be observed with fully fine-tuned models, where normalization yields diminishing benefits the more parameters we train. This behavior points to a trade-off between the use of our

Table 8: Tagger ablations with best hyperparameters $(h_\psi, d_{\text{MLP}})$. Best in bold, second-best underlined.

| $a$ | $N$ | $\phi$ | $\mathbf{e}_i^{tag}$ | ADE | CoNLL04 | SciERC | ERFGC | Mean |
|---|---|---|---|---|---|---|---|---|
| $(h_\psi, d_{\text{MLP}}) =$ | | | | (200, 100) | (400, 300) | (300, 300) | (400, 300) | |
| 1 | 0 | ■ | ■ | $0.541_{\pm 0.021}$ | $0.399_{\pm 0.024}$ | $0.147_{\pm 0.049}$ | $0.548_{\pm 0.010}$ | 0.515 |
| | 1 | ■ | ■ | $0.657_{\pm 0.011}$ | $0.556_{\pm 0.021}$ | $0.282_{\pm 0.009}$ | $0.676_{\pm 0.010}$ | |
| | 2 | ■ | ■ | $0.667_{\pm 0.011}$ | $0.573_{\pm 0.025}$ | $0.273_{\pm 0.010}$ | $0.694_{\pm 0.010}$ | |
| | 3 | ■ | ■ | $0.662_{\pm 0.027}$ | $0.562_{\pm 0.021}$ | $0.299_{\pm 0.023}$ | $0.705_{\pm 0.011}$ | |
| $\frac{1}{\sqrt{d}}$ | 0 | ■ | ■ | $0.567_{\pm 0.014}$ | $0.438_{\pm 0.033}$ | $0.181_{\pm 0.027}$ | $0.612_{\pm 0.008}$ | **0.541** |
| | 1 | ■ | ■ | $0.668_{\pm 0.017}$ | $\underline{0.597}_{\pm 0.015}$ | $0.299_{\pm 0.019}$ | $0.692_{\pm 0.009}$ | |
| | 2 | ■ | ■ | $0.676_{\pm 0.019}$ | $0.596_{\pm 0.014}$ | $0.312_{\pm 0.011}$ | $0.699_{\pm 0.009}$ | |
| | 3 | ■ | ■ | $\mathbf{0.686}_{\pm 0.025}$ | $\mathbf{0.602}_{\pm 0.017}$ | $0.320_{\pm 0.013}$ | $\mathbf{0.708}_{\pm 0.008}$ | |
| 1 | 0 | ■ | □ | $0.543_{\pm 0.013}$ | $0.402_{\pm 0.023}$ | $0.162_{\pm 0.019}$ | $0.554_{\pm 0.008}$ | 0.517 |
| | 1 | ■ | □ | $0.674_{\pm 0.011}$ | $0.527_{\pm 0.026}$ | $0.275_{\pm 0.013}$ | $0.677_{\pm 0.005}$ | |
| | 2 | ■ | □ | $0.672_{\pm 0.019}$ | $0.575_{\pm 0.009}$ | $0.293_{\pm 0.012}$ | $0.689_{\pm 0.011}$ | |
| | 3 | ■ | □ | $0.657_{\pm 0.027}$ | $0.583_{\pm 0.011}$ | $0.301_{\pm 0.012}$ | $0.697_{\pm 0.007}$ | |
| $\frac{1}{\sqrt{d}}$ | 0 | ■ | □ | $0.563_{\pm 0.013}$ | $0.443_{\pm 0.019}$ | $0.188_{\pm 0.006}$ | $0.612_{\pm 0.003}$ | 0.534 |
| | 1 | ■ | □ | $0.653_{\pm 0.023}$ | $0.565_{\pm 0.027}$ | $0.299_{\pm 0.020}$ | $0.687_{\pm 0.006}$ | |
| | 2 | ■ | □ | $0.672_{\pm 0.017}$ | $0.592_{\pm 0.020}$ | $0.307_{\pm 0.008}$ | $0.702_{\pm 0.005}$ | |
| | 3 | ■ | □ | $0.661_{\pm 0.022}$ | $0.593_{\pm 0.017}$ | $0.305_{\pm 0.014}$ | $\mathbf{0.708}_{\pm 0.007}$ | |
| 1 | 0 | □ | ■ | $0.550_{\pm 0.026}$ | $0.387_{\pm 0.030}$ | $0.157_{\pm 0.012}$ | $0.553_{\pm 0.006}$ | 0.514 |
| | 1 | □ | ■ | $0.667_{\pm 0.022}$ | $0.532_{\pm 0.036}$ | $0.273_{\pm 0.013}$ | $0.673_{\pm 0.006}$ | |
| | 2 | □ | ■ | $0.665_{\pm 0.021}$ | $0.565_{\pm 0.024}$ | $0.278_{\pm 0.027}$ | $0.694_{\pm 0.013}$ | |
| | 3 | □ | ■ | $0.676_{\pm 0.021}$ | $0.558_{\pm 0.051}$ | $0.288_{\pm 0.012}$ | $\underline{0.706}_{\pm 0.006}$ | |
| $\frac{1}{\sqrt{d}}$ | 0 | □ | ■ | $0.578_{\pm 0.022}$ | $0.437_{\pm 0.030}$ | $0.187_{\pm 0.014}$ | $0.611_{\pm 0.008}$ | 0.534 |
| | 1 | □ | ■ | $0.651_{\pm 0.032}$ | $0.559_{\pm 0.029}$ | $0.301_{\pm 0.007}$ | $0.684_{\pm 0.003}$ | |
| | 2 | □ | ■ | $0.673_{\pm 0.029}$ | $0.582_{\pm 0.017}$ | $0.310_{\pm 0.014}$ | $0.701_{\pm 0.008}$ | |
| | 3 | □ | ■ | $0.659_{\pm 0.019}$ | $0.586_{\pm 0.026}$ | $\mathbf{0.324}_{\pm 0.019}$ | $\mathbf{0.708}_{\pm 0.012}$ | |
| 1 | 0 | □ | □ | $0.547_{\pm 0.019}$ | $0.402_{\pm 0.033}$ | $0.156_{\pm 0.029}$ | $0.549_{\pm 0.015}$ | 0.516 |
| | 1 | □ | □ | $0.651_{\pm 0.017}$ | $0.552_{\pm 0.012}$ | $0.288_{\pm 0.008}$ | $0.680_{\pm 0.007}$ | |
| | 2 | □ | □ | $0.664_{\pm 0.031}$ | $0.566_{\pm 0.012}$ | $0.288_{\pm 0.017}$ | $0.693_{\pm 0.011}$ | |
| | 3 | □ | □ | $0.663_{\pm 0.008}$ | $0.561_{\pm 0.014}$ | $0.291_{\pm 0.012}$ | $0.703_{\pm 0.007}$ | |
| $\frac{1}{\sqrt{d}}$ | 0 | □ | □ | $0.566_{\pm 0.014}$ | $0.431_{\pm 0.015}$ | $0.182_{\pm 0.023}$ | $0.606_{\pm 0.012}$ | 0.534 |
| | 1 | □ | □ | $0.655_{\pm 0.015}$ | $0.567_{\pm 0.010}$ | $0.286_{\pm 0.018}$ | $0.678_{\pm 0.013}$ | |
| | 2 | □ | □ | $0.678_{\pm 0.014}$ | $0.591_{\pm 0.026}$ | $0.307_{\pm 0.007}$ | $0.702_{\pm 0.007}$ | |
| | 3 | □ | □ | $\underline{0.684}_{\pm 0.022}$ | $0.595_{\pm 0.017}$ | $\underline{0.321}_{\pm 0.016}$ | $0.698_{\pm 0.015}$ | |

technique and the amount of stacked BiLSTM layers, on a given dataset and at a given amount of training steps.

### D.3 MLP output dimension

As shown in Table 9, without normalization, increasing the output dimension of the two MLPs responsible for projecting edge representations leads to a decrease in performance. This is in contrast with the behavior observed when using normalization, where performance is essentially independent from the output dimension. This is in line with our claim that higher variance in the edge scores causes lower performance. Normalizing the scores reduces the variance, which makes performance stable even at high output dimensions. Once again, these results show the relationship between score variance and performance, with equivalent performance being achievable with fewer parameters.

### D.4 BiLSTM hidden size

As Table 10 shows, the hidden size of the BiLSTMs does not have a visible effect on performance. This is especially the case when applying score normalization, which produces smaller standard

Table 9: Performance in terms of $F_1$-measure when ablating over different output dimensions for the parser's MLPs ($\phi = \checkmark$, $\mathbf{e}_i^{tag} = \checkmark$). Best in bold.

| $a$ | $d_{\mathbf{MLP}}$ | **ADE** | **CoNLL04** | **SciERC** | **ERFGC** |
|---|---|---|---|---|---|
| $(N, h_\psi) =$ | | (3, 200) | (3, 400) | (3, 300) | (3, 400) |
| | 100 | $0.662_{\pm 0.027}$ | $0.579_{\pm 0.030}$ | $0.302_{\pm 0.018}$ | $0.709_{\pm 0.008}$ |
| 1 | 300 | $0.658_{\pm 0.018}$ | $0.562_{\pm 0.021}$ | $0.299_{\pm 0.023}$ | $0.705_{\pm 0.011}$ |
| | 500 | $0.657_{\pm 0.018}$ | $0.566_{\pm 0.019}$ | $0.281_{\pm 0.016}$ | $0.701_{\pm 0.009}$ |
| | 100 | $0.686_{\pm 0.025}$ | $0.586_{\pm 0.013}$ | $0.318_{\pm 0.017}$ | $\mathbf{0.712}_{\pm 0.008}$ |
| $\frac{1}{\sqrt{d}}$ | 300 | $0.680_{\pm 0.018}$ | $\mathbf{0.602}_{\pm 0.017}$ | $\mathbf{0.320}_{\pm 0.013}$ | $0.708_{\pm 0.008}$ |
| | 500 | $\mathbf{0.685}_{\pm 0.019}$ | $0.600_{\pm 0.015}$ | $0.311_{\pm 0.009}$ | $0.707_{\pm 0.004}$ |

Table 10: Performance in terms of $F_1$-measure when ablating over different hidden sizes for the parser's stacked BiLSTMs ($\phi = \checkmark$, $\mathbf{e}_i^{tag} = \checkmark$). Best in bold.

| $a$ | $h_\psi$ | **ADE** | **CoNLL04** | **SciERC** | **ERFGC** |
|---|---|---|---|---|---|
| $(N, d_{\mathbf{MLP}}) =$ | | (3, 100) | (3, 300) | (3, 300) | (3, 300) |
| | 100 | $0.682_{\pm 0.021}$ | $0.580_{\pm 0.031}$ | $0.291_{\pm 0.021}$ | $0.693_{\pm 0.011}$ |
| 1 | 200 | $0.662_{\pm 0.027}$ | $0.582_{\pm 0.006}$ | $0.286_{\pm 0.032}$ | $0.703_{\pm 0.007}$ |
| | 300 | $0.674_{\pm 0.029}$ | $0.585_{\pm 0.026}$ | $0.299_{\pm 0.023}$ | $0.703_{\pm 0.006}$ |
| | 400 | $0.663_{\pm 0.032}$ | $0.562_{\pm 0.021}$ | $0.289_{\pm 0.038}$ | $0.705_{\pm 0.011}$ |
| | 100 | $0.685_{\pm 0.020}$ | $0.600_{\pm 0.018}$ | $0.302_{\pm 0.013}$ | $0.698_{\pm 0.008}$ |
| $\frac{1}{\sqrt{d}}$ | 200 | $\mathbf{0.686}_{\pm 0.025}$ | $\mathbf{0.610}_{\pm 0.018}$ | $0.314_{\pm 0.019}$ | $0.702_{\pm 0.008}$ |
| | 300 | $0.678_{\pm 0.012}$ | $0.599_{\pm 0.012}$ | $\mathbf{0.320}_{\pm 0.013}$ | $\mathbf{0.711}_{\pm 0.005}$ |
| | 400 | $0.674_{\pm 0.011}$ | $0.602_{\pm 0.017}$ | $0.306_{\pm 0.016}$ | $0.708_{\pm 0.008}$ |

deviations. As a result, not only do models perform better with biaffine score normalization, but performance is also less dependent on the hidden size of the BiLSTM encoders. This supports our claim that score normalization is a useful technique to obtain the same performance with lower parameter counts, since the models tend to perform comparably, despite lower BiLSTM hidden sizes.

### D.5 LayerNorm and parameter initialization

In this section, we analyze the effects of using a Xavier normal initialization ($I_{par} \sim \mathcal{N}$)[7] for the weights of the two projections MLP$^{(edge-head)}$ and MLP$^{(edge-dept)}$, along with the edge biaffine layer $f^{(edge)}(\,\cdot\,; W_e)$. We also apply a LayerNorm function LN$_\psi$ [2] for each of the BiLSTM layers.

As reported in Table 11, on average the best results are obtained with the base setting, i.e., with $I_{par} \sim \mathcal{U}$ and LN$_\psi = \times$. Using LayerNorm can provide a performance boost for some datasets, especially with uniform initialization. However, it makes performance unstable for some. As a matter of fact, when using LayerNorm with three BiLSTM layers on CoNLL04 and SciERC, the model often breaks and is not able to converge. Based on average performance, then, the best models are obtained by scaling biaffine scores ($a = 1/\sqrt{d}$) and initializing the parser with a uniform weight distribution, without any LayerNorm in between the stacked BiLSTM layers.

### D.6 Full fine-tuning and sample efficiency

In this section, we present the results obtained when fine-tuning BERT$_{base}$, DeBERTa$_{base}$, BERT$_{large}$, and DeBERTa$_{large}$ over $10k$ steps. In our previous settings, we only used a frozen BERT$_{base}$, whose last hidden states we fed as input to the tagger and parser. This evaluation allows us to test the effectiveness of our approach in a setting in which the number of learnable parameters is unconstrained. In this experiment, we use different learning rates for the base models ($\eta = 1 \times 10^{-4}$) and the large models ($\eta = 3 \times 10^{-5}$) and we apply gradient norm clipping with $\|\nabla\|_{max} = 1.0$. In

---

[7]`https://docs.pytorch.org/docs/stable/nn.init.html#torch.nn.init.xavier_normal_`

Table 11: Ablation over the best hyperparameters $(h_\psi, d_{\mathrm{MLP}})$ and $\phi = \checkmark$ and $\mathbf{e}_i^{tag} = \checkmark$, using LayerNorm layers $\mathrm{LN}_\psi \in \{\checkmark, \times\}$ and Xavier uniform/normal initialization $I_{par} \sim \{\mathcal{U}, \mathcal{N}\}$ for the parser. The first quarter of the table is equivalent to Table 3. Best in bold, second-best underlined.

| $I_{par}$ | $\mathrm{LN}_\psi$ | $a$ | $N$ | **ADE** | **CoNLL04** | **SciERC** | **ERFGC** | **Mean** |
|---|---|---|---|---|---|---|---|---|
| $(h_\psi, d_{\mathrm{MLP}}) =$ | | | | (200, 100) | (400, 300) | (300, 300) | (400, 300) | |
| | □ | | 0 | $0.541\ _{\pm 0.021}$ | $0.399\ _{\pm 0.024}$ | $0.147\ _{\pm 0.049}$ | $0.548\ _{\pm 0.010}$ | |
| | □ | 1 | 3 | $0.662\ _{\pm 0.027}$ | $0.562\ _{\pm 0.021}$ | $0.299\ _{\pm 0.023}$ | $0.705\ _{\pm 0.011}$ | 0.515 |
| | □ | | 1 | $0.657\ _{\pm 0.011}$ | $0.556\ _{\pm 0.021}$ | $0.282\ _{\pm 0.009}$ | $0.676\ _{\pm 0.010}$ | |
| | □ | | 2 | $0.667\ _{\pm 0.011}$ | $0.573\ _{\pm 0.025}$ | $0.273\ _{\pm 0.010}$ | $0.694\ _{\pm 0.010}$ | |
| | □ | | 0 | $0.567\ _{\pm 0.014}$ | $0.438\ _{\pm 0.033}$ | $0.181\ _{\pm 0.027}$ | $0.612\ _{\pm 0.008}$ | |
| | □ | $\frac{1}{\sqrt{d}}$ | 1 | $0.668\ _{\pm 0.017}$ | $0.597\ _{\pm 0.015}$ | $0.299\ _{\pm 0.019}$ | $0.692\ _{\pm 0.009}$ | **0.541** |
| | □ | | 2 | $0.676\ _{\pm 0.019}$ | $0.596\ _{\pm 0.014}$ | $0.312\ _{\pm 0.011}$ | $0.699\ _{\pm 0.009}$ | |
| $\mathcal{U}$ | □ | | 3 | $0.686\ _{\pm 0.025}$ | $0.602\ _{\pm 0.017}$ | $\mathbf{0.320}\ _{\pm 0.013}$ | $0.708\ _{\pm 0.008}$ | |
| | ■ | | 0 | $0.545\ _{\pm 0.018}$ | $0.399\ _{\pm 0.024}$ | $0.151\ _{\pm 0.014}$ | $0.548\ _{\pm 0.010}$ | |
| | ■ | 1 | 1 | $0.680\ _{\pm 0.014}$ | $0.554\ _{\pm 0.042}$ | $0.265\ _{\pm 0.028}$ | $0.686\ _{\pm 0.007}$ | 0.468 |
| | ■ | | 2 | $0.679\ _{\pm 0.011}$ | $0.578\ _{\pm 0.033}$ | $0.260\ _{\pm 0.020}$ | $\mathbf{0.715}\ _{\pm 0.012}$ | |
| | ■ | | 3 | $0.680\ _{\pm 0.013}$ | $0.117\ _{\pm 0.261}$ | $0.060\ _{\pm 0.133}$ | $0.569\ _{\pm 0.318}$ | |
| | ■ | | 0 | $0.567\ _{\pm 0.014}$ | $0.433\ _{\pm 0.029}$ | $0.181\ _{\pm 0.027}$ | $0.614\ _{\pm 0.004}$ | |
| | ■ | $\frac{1}{\sqrt{d}}$ | 1 | $0.664\ _{\pm 0.017}$ | $0.564\ _{\pm 0.037}$ | $0.282\ _{\pm 0.029}$ | $0.686\ _{\pm 0.004}$ | 0.503 |
| | ■ | | 2 | $\mathbf{0.697}\ _{\pm 0.022}$ | $\mathbf{0.623}\ _{\pm 0.019}$ | $0.300\ _{\pm 0.042}$ | $0.702\ _{\pm 0.011}$ | |
| | ■ | | 3 | $0.675\ _{\pm 0.019}$ | $0.239\ _{\pm 0.327}$ | $0.111\ _{\pm 0.152}$ | $0.703\ _{\pm 0.006}$ | |
| | □ | | 0 | $0.545\ _{\pm 0.017}$ | $0.415\ _{\pm 0.014}$ | $0.155\ _{\pm 0.019}$ | $0.558\ _{\pm 0.009}$ | |
| | □ | 1 | 1 | $0.667\ _{\pm 0.014}$ | $0.543\ _{\pm 0.017}$ | $0.275\ _{\pm 0.020}$ | $0.680\ _{\pm 0.015}$ | 0.520 |
| | □ | | 2 | $0.674\ _{\pm 0.025}$ | $0.576\ _{\pm 0.023}$ | $0.272\ _{\pm 0.014}$ | $0.699\ _{\pm 0.006}$ | |
| | □ | | 3 | $0.672\ _{\pm 0.028}$ | $0.580\ _{\pm 0.022}$ | $0.297\ _{\pm 0.019}$ | $0.705\ _{\pm 0.006}$ | |
| | □ | | 0 | $0.570\ _{\pm 0.013}$ | $0.454\ _{\pm 0.006}$ | $0.181\ _{\pm 0.023}$ | $0.613\ _{\pm 0.007}$ | |
| | □ | $\frac{1}{\sqrt{d}}$ | 1 | $0.671\ _{\pm 0.015}$ | $0.578\ _{\pm 0.031}$ | $0.299\ _{\pm 0.030}$ | $0.685\ _{\pm 0.010}$ | $\underline{0.538}$ |
| $\mathcal{N}$ | □ | | 2 | $0.671\ _{\pm 0.031}$ | $0.593\ _{\pm 0.016}$ | $0.301\ _{\pm 0.019}$ | $0.704\ _{\pm 0.007}$ | |
| | □ | | 3 | $0.681\ _{\pm 0.024}$ | $0.590\ _{\pm 0.014}$ | $\underline{0.315}\ _{\pm 0.009}$ | $0.702\ _{\pm 0.011}$ | |
| | ■ | | 0 | $0.545\ _{\pm 0.017}$ | $0.415\ _{\pm 0.014}$ | $0.155\ _{\pm 0.019}$ | $0.558\ _{\pm 0.009}$ | |
| | ■ | 1 | 1 | $0.679\ _{\pm 0.012}$ | $0.582\ _{\pm 0.012}$ | $0.259\ _{\pm 0.010}$ | $0.680\ _{\pm 0.016}$ | 0.469 |
| | ■ | | 2 | $0.668\ _{\pm 0.015}$ | $0.580\ _{\pm 0.041}$ | $0.284\ _{\pm 0.024}$ | $\underline{0.711}\ _{\pm 0.005}$ | |
| | ■ | | 3 | $0.679\ _{\pm 0.018}$ | $0.000\ _{\pm 0.000}$ | $0.000\ _{\pm 0.000}$ | $\underline{0.711}\ _{\pm 0.009}$ | |
| | ■ | | 0 | $0.570\ _{\pm 0.013}$ | $0.454\ _{\pm 0.006}$ | $0.181\ _{\pm 0.023}$ | $0.613\ _{\pm 0.007}$ | |
| | ■ | $\frac{1}{\sqrt{d}}$ | 1 | $0.675\ _{\pm 0.019}$ | $0.578\ _{\pm 0.022}$ | $0.280\ _{\pm 0.022}$ | $0.682\ _{\pm 0.006}$ | 0.512 |
| | ■ | | 2 | $\underline{0.687}\ _{\pm 0.021}$ | $\underline{0.609}\ _{\pm 0.012}$ | $0.308\ _{\pm 0.012}$ | $0.702\ _{\pm 0.011}$ | |
| | ■ | | 3 | $0.678\ _{\pm 0.009}$ | $0.476\ _{\pm 0.266}$ | $0.000\ _{\pm 0.000}$ | $0.701\ _{\pm 0.006}$ | |

Table 12: Results for the fully fine-tuned models ($h_\psi = 400$, $h_{out} = 500$, $\phi = \checkmark$, and $\mathbf{e}_i^{tag} = \checkmark$).

| Model | $a$ | ADE | CoNLL04 | SciERC | ERFGC | Mean |
|---|---|---|---|---|---|---|
| BERT$_{base}$ | 1 | 0.748 $_{\pm 0.028}$ | 0.613 $_{\pm 0.019}$ | 0.414 $_{\pm 0.011}$ | 0.726 $_{\pm 0.006}$ | 0.321 |
| | $\frac{1}{\sqrt{d}}$ | 0.731 $_{\pm 0.025}$ | 0.629 $_{\pm 0.022}$ | 0.412 $_{\pm 0.016}$ | 0.726 $_{\pm 0.006}$ | 0.321 |
| BERT$_{large}$ | 1 | 0.748 $_{\pm 0.016}$ | 0.700 $_{\pm 0.019}$ | 0.473 $_{\pm 0.011}$ | 0.750 $_{\pm 0.006}$ | 0.340 |
| | $\frac{1}{\sqrt{d}}$ | 0.777 $_{\pm 0.011}$ | 0.697 $_{\pm 0.014}$ | 0.446 $_{\pm 0.025}$ | 0.748 $_{\pm 0.010}$ | 0.341 |
| DeBERTa$_{base}$ | 1 | 0.754 $_{\pm 0.013}$ | 0.674 $_{\pm 0.014}$ | 0.429 $_{\pm 0.015}$ | 0.751 $_{\pm 0.002}$ | 0.332 |
| | $\frac{1}{\sqrt{d}}$ | 0.761 $_{\pm 0.021}$ | 0.700 $_{\pm 0.013}$ | 0.425 $_{\pm 0.019}$ | 0.751 $_{\pm 0.010}$ | 0.338 |
| DeBERTa$_{large}$ | 1 | 0.786 $_{\pm 0.010}$ | 0.739 $_{\pm 0.016}$ | 0.478 $_{\pm 0.019}$ | 0.763 $_{\pm 0.006}$ | 0.352 |
| | $\frac{1}{\sqrt{d}}$ | 0.794 $_{\pm 0.015}$ | 0.747 $_{\pm 0.013}$ | 0.476 $_{\pm 0.010}$ | 0.763 $_{\pm 0.007}$ | 0.353 |

the case of the large models, we also use a cosine schedule with warm-up over 6% of the steps. We adopt these measures because during early trials we experienced sudden mid-run gradient explosions. In this case, we do not use any downstream BiLSTMs and only vary whether we use biaffine score normalization in the parser.

Table 12 reports the performance in terms of micro-averaged $F_1$-measure for the best models, picked based on top validation performance, evaluated every 100 steps. As the table shows, normalizing the scores generally does not increase top performance when fully fine-tuning these models. This is in line with our theoretical results, since tuning all of their $\sim 100 - 400M$ parameters can compensate for the lack of biaffine score normalization.

In order to have a stronger indication of whether score normalization also works in this setting, we study the performance on the test set versus the amount of training steps. Figure 7 and Figure 8 respectively visualize the performance of the base and large models on the test set in terms of micro-averaged $F_1$-measure over $10k$ training steps. We still use early stopping, which is why some of the series are cut short before reaching $10k$ steps.

As Figure 7 shows, a one-tailed Wilcoxon signed-rank test finds the difference in performance throughout the training to be significantly higher when normalizing scores. Since the average test performance of the models is similar with and without normalization, the gap between the two curves being statistically significant indicates faster convergence with normalization.

For both BERT$_{base}$ and DeBERTa$_{base}$, the difference in convergence speed and performance is statistically significant on ADE, CoNLL04, and SciERC. The same is true for BERT$_{large}$ only on ADE. However, the effect is statistically significant with DeBERTa$_{large}$ for all datasets. This indicates our score normalization approach can produce positive effects in terms of sample efficiency also when fine-tuning models with hundreds of millions of trainable parameters.

## E  Full main-setting results

Table 13 reports the main-setting results for the best hyperparameter combinations for all three tasks of predicting tags, edges, and relations of the semantic dependency graphs. The first part of Table 13 is identical to the bottom of Table 3. Since we already discussed the performance for labeled edges in Section 5, we forgo discussing them again in this appendix.

As regards unlabeled edges, using score normalization improves mean performance for all datasets, similarly to relations. This is expected, as unlabeled edges directly influence the prediction of the edge labels, together with entity class prediction. As we have already observed for relations, the performance boost provided by normalization is particularly great for SciERC at $N = 0$.

It is also interesting to notice that for CoNLL04, the performance for the unlabeled edge prediction task is only slightly higher than that of the relations. In fact, as regards ADE, the opposite is true, with the labeled performance seemingly being higher. Due to the high variances, it would therefore seem that for these two datasets there is essentially no difference in difficulty between the labeled and unlabeled edge prediction tasks. In the case of ADE, this makes perfect sense, since there is

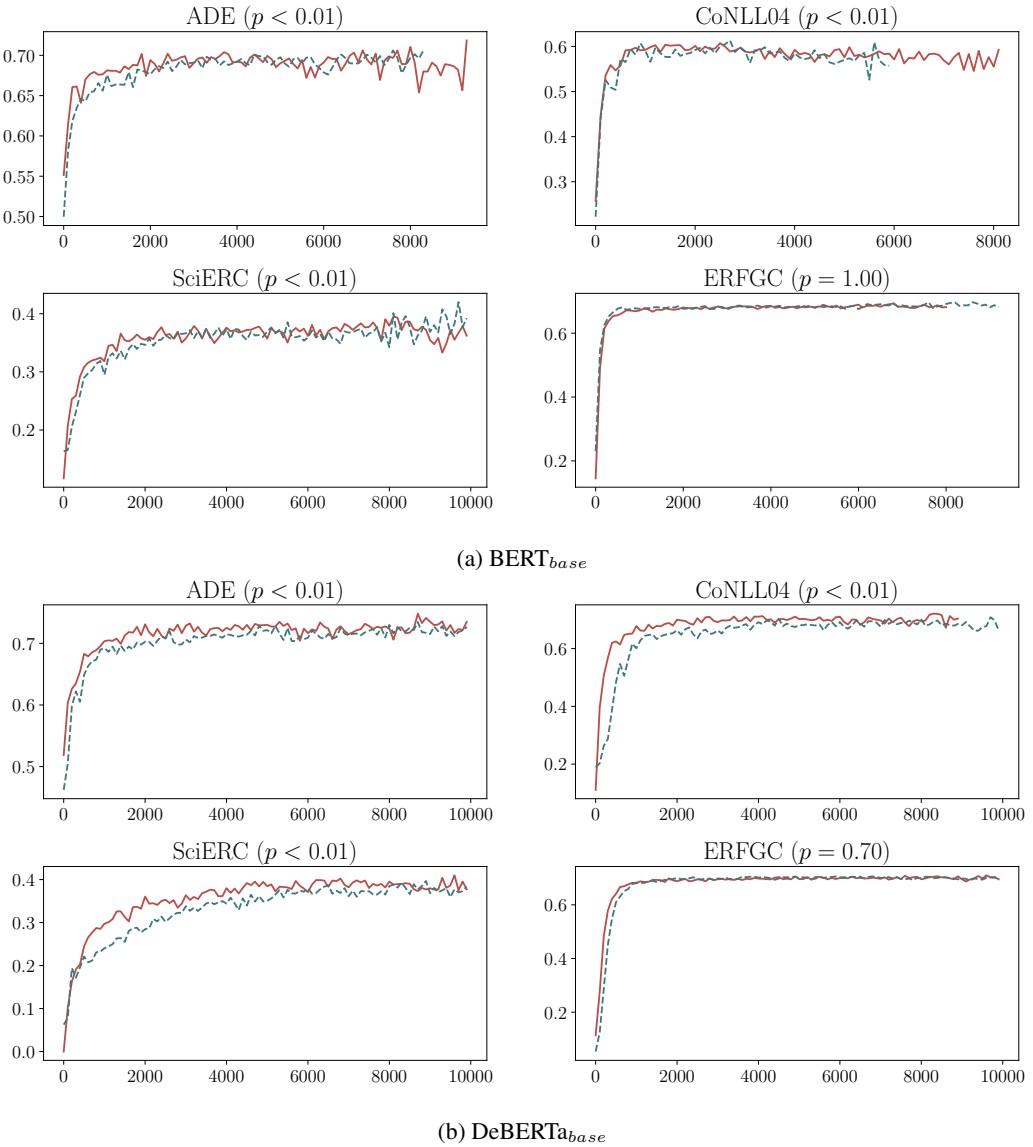

Figure 7: Performance in terms of $F_1$-measure vs the number of training steps for the base models on the SemDP datasets. Red = norm; blue = raw. The $p$-values refer to the performance being greater with score normalization (one-tailed Wilcoxon signed-rank test).

only one possible relation. For CoNLL04, the amount of relation types is rather limited as well; therefore, it is sensible for the performance to also be similar in this case. Indeed, when looking at SciERC and ERFGC, the performance gap between labeled and unlabeled tasks is considerable. This is a strong indication that CoNLL04 is thus found somewhere in the middle, where the number of possible relations only slightly affects labeled performance.

As regards enEWT and SciDTB, the performance on the prediction of edges (UAS) and relations (LAS) is also rather similar. Since in this case the predictions involve syntactic rather than semantic relations, the similar performance hints at relations being easier to predict in SynDP than in SemDP.

In the tagging task, very high standard deviations can be observed. In the case of ADE and CoNLL04, this could be caused by our choice of $\lambda_1 = 0.1$ being too low a value. Regarding SciERC, as already mentioned, most of the entities in the validation and test sets are not found in the training set. This means it is not possible to train the model to recognize them, which results in the abysmal tagging performance reported in Table 13. This makes it challenging to train good tag embeddings which can

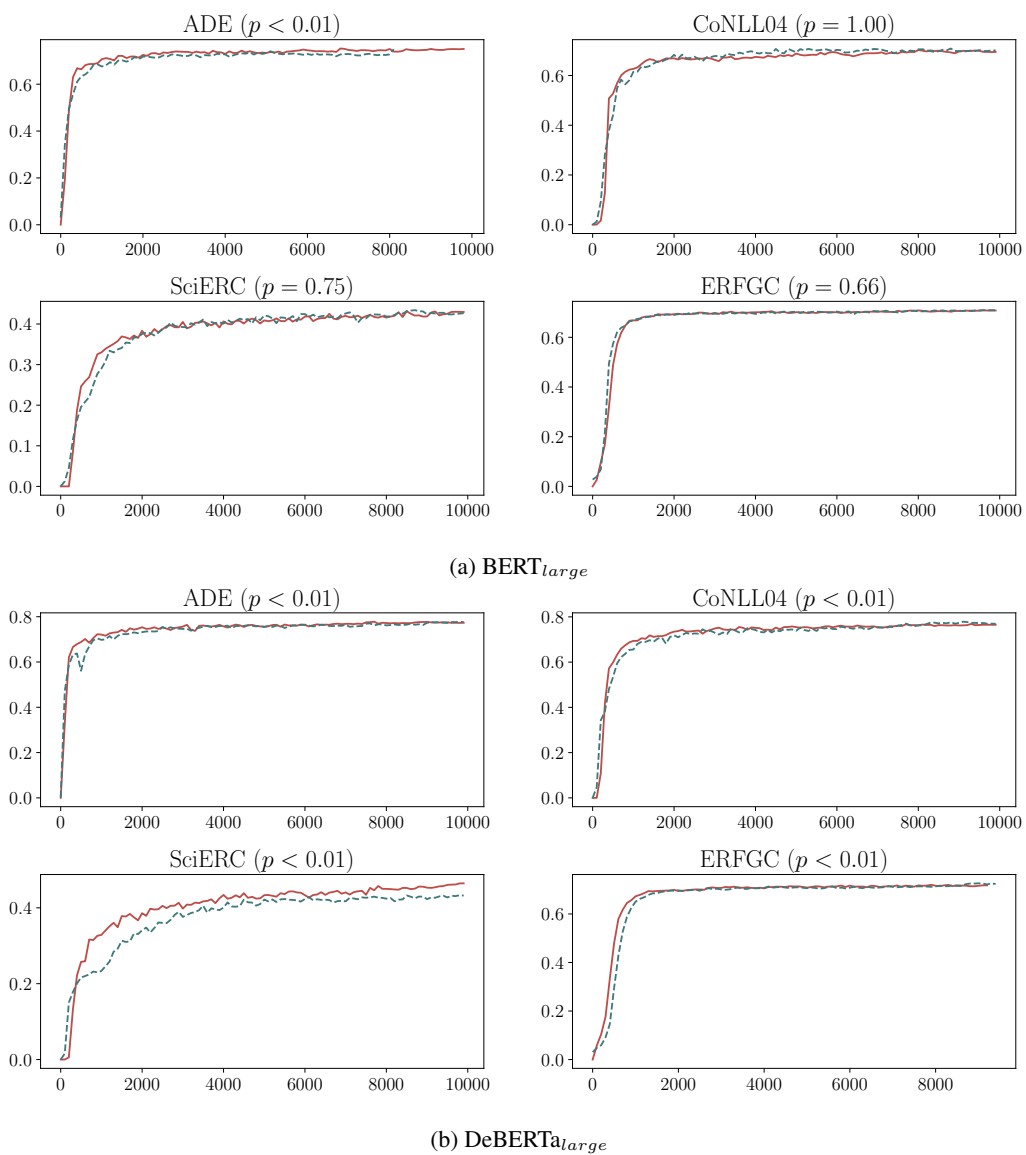

Figure 8: Performance in terms of $F_1$-measure vs the number of training steps for the large models on the SemDP datasets. Red = norm; blue = raw. The $p$-values refer to the performance being greater with score normalization (one-tailed Wilcoxon signed-rank test).

help inform the edge and relation prediction tasks. Surprisingly, we notice that for CoNLL04 and ERFGC the tagging performance is also seemingly higher with score normalization. However, the overlaps of the standard deviations are too high to be able to make any claims on the matter.

## F  Computational resources

We ran all of our experiments on a cluster of NVIDIA H100 (96GB of VRAM) and NVIDIA L40 (48GB of VRAM) GPUs, one run per single GPU. When freezing the $\text{BERT}_{base}$ encoder and only training the BiLSTMs and the classifiers, each training and evaluation run took ~5-7 minutes, depending on the number of BiLSTM layers. For the main setting, finding the best hyperparameters involved training and testing 6,120 models, for a total of ~600 GPU hours. In the full fine-tuning setting (Appendix D.6), training and evaluation took ~1 hour for each of the 40 base models and ~2-3 hours for each of the 40 large models, for an additional ~140 GPU hours.

Table 13: Micro-$F_1$ (SemDP) and LAS (SynDP) on all tasks ($\phi = \checkmark$, $\mathbf{e}_i^{tag} = \checkmark$). Best in bold.

| Metric | $a$ | $N$ | ADE | CoNLL04 | SciERC | ERFGC | enEWT | SciDTB |
|---|---|---|---|---|---|---|---|---|
| $(h_\psi, d_{\mathrm{MLP}}) =$ | | | (200, 100) | (400, 300) | (300, 300) | (400, 300) | (400, 500) | (400, 500) |
| rels. | 1 | 0 | $0.541_{\pm 0.021}$ | $0.399_{\pm 0.024}$ | $0.147_{\pm 0.049}$ | $0.548_{\pm 0.010}$ | $0.559_{\pm 0.005}$ | $0.729_{\pm 0.004}$ |
| | | 1 | $0.657_{\pm 0.011}$ | $0.556_{\pm 0.021}$ | $0.282_{\pm 0.009}$ | $0.676_{\pm 0.010}$ | $0.771_{\pm 0.006}$ | $0.892_{\pm 0.002}$ |
| | | 2 | $0.667_{\pm 0.011}$ | $0.573_{\pm 0.025}$ | $0.273_{\pm 0.010}$ | $0.694_{\pm 0.010}$ | $0.796_{\pm 0.006}$ | $0.910_{\pm 0.002}$ |
| | | 3 | $0.662_{\pm 0.027}$ | $0.562_{\pm 0.021}$ | $0.299_{\pm 0.023}$ | $0.705_{\pm 0.011}$ | $0.804_{\pm 0.006}$ | $0.915_{\pm 0.002}$ |
| | $\frac{1}{\sqrt{d}}$ | 0 | $0.567_{\pm 0.014}$ | $0.438_{\pm 0.033}$ | $0.181_{\pm 0.027}$ | $0.612_{\pm 0.008}$ | $0.646_{\pm 0.002}$ | $0.796_{\pm 0.002}$ |
| | | 1 | $0.668_{\pm 0.017}$ | $0.597_{\pm 0.015}$ | $0.299_{\pm 0.019}$ | $0.692_{\pm 0.009}$ | $0.789_{\pm 0.003}$ | $0.904_{\pm 0.002}$ |
| | | 2 | $0.676_{\pm 0.019}$ | $0.596_{\pm 0.014}$ | $0.312_{\pm 0.011}$ | $0.699_{\pm 0.009}$ | $0.805_{\pm 0.003}$ | $0.916_{\pm 0.002}$ |
| | | 3 | $\mathbf{0.686}_{\pm 0.025}$ | $\mathbf{0.602}_{\pm 0.017}$ | $\mathbf{0.320}_{\pm 0.013}$ | $\mathbf{0.708}_{\pm 0.008}$ | $\mathbf{0.807}_{\pm 0.005}$ | $\mathbf{0.919}_{\pm 0.001}$ |
| edges | 1 | 0 | $0.536_{\pm 0.009}$ | $0.415_{\pm 0.020}$ | $0.173_{\pm 0.050}$ | $0.601_{\pm 0.013}$ | $0.589_{\pm 0.006}$ | $0.745_{\pm 0.005}$ |
| | | 1 | $0.652_{\pm 0.009}$ | $0.566_{\pm 0.022}$ | $0.321_{\pm 0.009}$ | $0.744_{\pm 0.013}$ | $0.793_{\pm 0.006}$ | $0.901_{\pm 0.002}$ |
| | | 2 | $0.657_{\pm 0.021}$ | $0.586_{\pm 0.017}$ | $0.322_{\pm 0.017}$ | $0.769_{\pm 0.014}$ | $0.819_{\pm 0.006}$ | $0.919_{\pm 0.002}$ |
| | | 3 | $0.649_{\pm 0.027}$ | $0.578_{\pm 0.011}$ | $0.355_{\pm 0.028}$ | $0.782_{\pm 0.007}$ | $0.827_{\pm 0.006}$ | $0.924_{\pm 0.002}$ |
| | $\frac{1}{\sqrt{d}}$ | 0 | $0.549_{\pm 0.014}$ | $0.453_{\pm 0.032}$ | $0.203_{\pm 0.030}$ | $0.674_{\pm 0.007}$ | $0.682_{\pm 0.003}$ | $0.815_{\pm 0.001}$ |
| | | 1 | $0.654_{\pm 0.020}$ | $0.598_{\pm 0.021}$ | $0.351_{\pm 0.019}$ | $0.762_{\pm 0.009}$ | $0.810_{\pm 0.003}$ | $0.913_{\pm 0.002}$ |
| | | 2 | $0.660_{\pm 0.015}$ | $0.600_{\pm 0.008}$ | $0.367_{\pm 0.015}$ | $0.775_{\pm 0.008}$ | $0.827_{\pm 0.003}$ | $0.925_{\pm 0.002}$ |
| | | 3 | $\mathbf{0.666}_{\pm 0.028}$ | $\mathbf{0.610}_{\pm 0.022}$ | $\mathbf{0.377}_{\pm 0.021}$ | $\mathbf{0.786}_{\pm 0.004}$ | $\mathbf{0.829}_{\pm 0.004}$ | $\mathbf{0.928}_{\pm 0.002}$ |
| tags | 1 | 0 | $0.615_{\pm 0.115}$ | $0.285_{\pm 0.150}$ | $0.014_{\pm 0.015}$ | $0.662_{\pm 0.073}$ | | |
| | | 1 | $0.665_{\pm 0.147}$ | $0.671_{\pm 0.018}$ | $0.049_{\pm 0.014}$ | $0.794_{\pm 0.057}$ | | |
| | | 2 | $0.746_{\pm 0.011}$ | $0.648_{\pm 0.074}$ | $0.060_{\pm 0.012}$ | $0.838_{\pm 0.016}$ | | |
| | | 3 | $\mathbf{0.768}_{\pm 0.022}$ | $0.669_{\pm 0.033}$ | $\mathbf{0.062}_{\pm 0.005}$ | $0.853_{\pm 0.013}$ | | |
| | $\frac{1}{\sqrt{d}}$ | 0 | $0.604_{\pm 0.134}$ | $0.464_{\pm 0.115}$ | $0.046_{\pm 0.005}$ | $0.735_{\pm 0.066}$ | | |
| | | 1 | $0.734_{\pm 0.048}$ | $0.702_{\pm 0.031}$ | $0.058_{\pm 0.014}$ | $0.857_{\pm 0.010}$ | | |
| | | 2 | $0.724_{\pm 0.064}$ | $0.688_{\pm 0.080}$ | $0.060_{\pm 0.007}$ | $\mathbf{0.864}_{\pm 0.018}$ | | |
| | | 3 | $0.762_{\pm 0.017}$ | $\mathbf{0.690}_{\pm 0.062}$ | $0.057_{\pm 0.012}$ | $0.850_{\pm 0.022}$ | | |

