# OpenReview forum: "Dependency Parsing is More Parameter-Efficient with Normalization"
_NeurIPS.cc/2025/Conference — NeurIPS 2025 poster_

### Official Review · Reviewer_mCBU · 2025-06-21

**Clarity:** 3
**Significance:** 2
**Originality:** 3
**Rating:** 3
**Confidence:** 4

**Summary:**

This work studies the effect of normalization on the scores produced by biaffine functions in dependency parsing tasks. The authors demonstrate, both theoretically and empirically, a lack of normalization necessarily results in overparameterized parser models, where the extra parameters compensate for the sharp softmax outputs produced by high variance inputs to the biaffine scoring function. Experimental results show that similar or better performance can be obtained by reducing the amount of trained BiLSTM parameters by as much as 85%.

**Questions:**

See the weaknesses.

**Ethical Concerns:**

["NO or VERY MINOR ethics concerns only"]

**Final Justification:**

First, I do not think the contribution well-suitable for NeurIPS. Second,  even in the community of *ACL, the contribution is not strong enough. Might be better for a workshop oriented to syntax parsing.

**Limitations:**

yes

**Quality:**

2

**Strengths And Weaknesses:**

Strengths:
1. This paper explores a problem that is overlooked. Biaffine function is a standard implement, and few people pay attention to it and its relationship with parameter quantity.
2. Comprehensive experiments.

Weaknesses:
1. Although this research question is very detailed and easily overlooked, I don't think this contribution meets the acceptance bar of NeurIPS.
2. Are there consistent experimental results on other language datasets? Experimental conclusion should be verified on more languages.
3. What are experimental results if add a Normal Layer?

---

> ### Author Rebuttal · Authors · 2025-07-30
>
> Q.1: Although this research question is very detailed and easily overlooked, I don't think this contribution meets the acceptance bar of NeurIPS.
>
> A.1: We agree with the reviewer that the topic of our contribution is indeed overlooked. Thus, our work is novel in terms of improving dependency parsing as an application. Furthermore, our contribution involves analyzing normalization phenomena in architectures used broadly in graph inference tasks (i.e., modeling edges as fully connected weighted graphs via attention mechanisms, such as biaffine transformations, see [1,2,3] for reference). To show this, we have repeated the experiment using Graph Attention Networks (GAT), using a stack of biaffine/GAT pairs to encode higher-order dependencies before the final biaffine layer. The results with this new architecture are consistent, as shown in the table below. Moreover, we have expanded the experiments to six multilingual settings as reported in our answer to the reviewer's next comment.
>
> |   norm |   $L_\phi$ | CoNLL04           | ADE               | SciERC            | enEWT             | SciDTB             | ERFGC             |
> |-------:|----:|:------------------|:------------------|:------------------|:------------------|:------------------|:------------------|
> |      0 |   0 | 0.526 $\pm 0.046$ | 0.517 $\pm 0.038$ | 0.123 $\pm 0.050$ | 0.610 $\pm 0.009$ | 0.727 $\pm 0.006$ | 0.536 $\pm 0.013$ |
> |      0 |   1 | 0.493 $\pm 0.037$ | 0.509 $\pm 0.022$ | 0.039 $\pm 0.020$ | 0.583 $\pm 0.011$ | 0.731 $\pm 0.009$ | 0.599 $\pm 0.008$ |
> |      0 |   2 | 0.485 $\pm 0.033$ | 0.509 $\pm 0.025$ | 0.029 $\pm 0.041$ | 0.540 $\pm 0.012$ | 0.710 $\pm 0.013$ | 0.611 $\pm 0.015$ |
> |      0 |   3 | 0.481 $\pm 0.037$ | 0.487 $\pm 0.040$ | 0.000 $\pm 0.000$ | 0.511 $\pm 0.015$ | 0.700 $\pm 0.010$ | 0.585 $\pm 0.009$ |
> |      1 |   0 | 0.574 $\pm 0.028$ | 0.556 $\pm 0.037$ | 0.156 $\pm 0.031$ | 0.671 $\pm 0.007$ | 0.778 $\pm 0.006$ | 0.607 $\pm 0.009$ |
> |      1 |   1 | 0.550 $\pm 0.036$ | 0.587 $\pm 0.015$ | 0.148 $\pm 0.028$ | 0.696 $\pm 0.005$ | 0.809 $\pm 0.008$ | 0.639 $\pm 0.007$ |
> |      1 |   2 | 0.563 $\pm 0.029$ | 0.534 $\pm 0.023$ | 0.128 $\pm 0.029$ | 0.657 $\pm 0.007$ | 0.795 $\pm 0.008$ | 0.637 $\pm 0.007$ |
> |      1 |   3 | 0.514 $\pm 0.026$ | 0.543 $\pm 0.024$ | 0.071 $\pm 0.021$ | 0.596 $\pm 0.010$ | 0.770 $\pm 0.007$ | 0.616 $\pm 0.006$ |
>
> Score normalization improves performance also when using GATs, showing that our method generalizes to other architectures, besides Transformers and BiLSTMS. We have included these results to the paper.
>
> In addition to our technical contribution, we believe that our work contributes to the broad NeurIPS community, by suggesting the importance of peculiar settings such as normalization in the training of neural network models. Moreover, we would like to respectfully point out that the NeurIPS guideline of acceptance suggests a work with "high impact on at least one sub-area of AI" (per grade 5), meaning that broad impact is not strictly required for acceptance. We appreciate the reviewer's consideration in this regard.
>
> *[1] Kazi et al. "Differentiable Graph Module (DGM) for Graph Convolutional Networks". 2023.
>
> *[2] P. Velickovic, G. Cucurull, A. Casanova, A. Romero, P. Lio, and Y. Bengio, "Graph attention networks". 2017.
>
> *[3] J. Zhang, X. Shi, J. Xie, H. Ma, I. King, and D.-Y. Yeung, "GAAN: Gated attention networks for learning on large and spatiotemporal graphs". 2018.
>
> Q.2: Are there consistent experimental results on other language datasets? Experimental conclusion should be verified on more languages.
>
> A.2: We thank the reviewer for pointing this out. We performed multilingual experiments using six non-English UD datasets (UD_Arabic-PADT, UD_Chinese-GSD, UD_Italian-ISDT, UD_Japanese-GSD, UD_Spanish-AnCora, UD_Wolof-WTB) and verified that our finding is consistent, as shown in the table below.
>
> | score norm | $L_\psi$ | UD_Arabic-PADT    | UD_Chinese-GSD    | UD_Italian-ISDT   | UD_Japanese-GSD   | UD_Spanish-AnCora   | UD_Wolof-WTB      |
> |:---|:----|:------------------|:------------------|:------------------|:------------------|:--------------------|:------------------|
> | $\times$ | 0 | 0.538 $\pm\scriptstyle 0.005$ | 0.395 $\pm\scriptstyle 0.007$ | 0.563 $\pm\scriptstyle 0.006$ | 0.493 $\pm\scriptstyle 0.010$ | 0.554 $\pm\scriptstyle 0.004$   | 0.252 $\pm\scriptstyle 0.007$ |
> | $\times$ | 1 | 0.723 $\pm\scriptstyle 0.008$ | 0.653 $\pm\scriptstyle 0.007$ | 0.792 $\pm\scriptstyle 0.003$ | 0.812 $\pm\scriptstyle 0.003$ | 0.775 $\pm\scriptstyle 0.003$   | 0.525 $\pm\scriptstyle 0.006$ |
> | $\times$ | 2 | 0.745 $\pm\scriptstyle 0.004$ | 0.710 $\pm\scriptstyle 0.005$ | 0.826 $\pm\scriptstyle 0.003$ | 0.844 $\pm\scriptstyle 0.005$ | 0.807 $\pm\scriptstyle 0.002$   | 0.587 $\pm\scriptstyle 0.007$ |
> | $\times$ | 3 | 0.748 $\pm\scriptstyle 0.004$ | 0.717 $\pm\scriptstyle 0.007$ | 0.832 $\pm\scriptstyle 0.002$ | 0.849 $\pm\scriptstyle 0.003$ | 0.808 $\pm\scriptstyle 0.002$   | 0.614 $\pm\scriptstyle 0.005$ |
> | $\checkmark$ | 0 | 0.609 $\pm\scriptstyle 0.003$ | 0.479 $\pm\scriptstyle 0.007$ | 0.633 $\pm\scriptstyle 0.002$ | 0.585 $\pm\scriptstyle 0.005$ | 0.620 $\pm\scriptstyle 0.002$   | 0.305 $\pm\scriptstyle 0.005$ |
> | $\checkmark$ | 1 | 0.737 $\pm\scriptstyle 0.007$ | 0.693 $\pm\scriptstyle 0.003$ | 0.820 $\pm\scriptstyle 0.004$ | 0.838 $\pm\scriptstyle 0.004$ | 0.801 $\pm\scriptstyle 0.002$   | 0.556 $\pm\scriptstyle 0.003$ |
> | $\checkmark$ | 2 | 0.758 $\pm\scriptstyle 0.003$ | 0.736 $\pm\scriptstyle 0.005$ | 0.842 $\pm\scriptstyle 0.004$ | 0.859 $\pm\scriptstyle 0.004$ | 0.822 $\pm\scriptstyle 0.001$   | 0.613 $\pm\scriptstyle 0.008$ |
> | $\checkmark$ | 3 | 0.759 $\pm\scriptstyle 0.005$ | 0.742 $\pm\scriptstyle 0.006$ | 0.845 $\pm\scriptstyle 0.002$ | 0.859 $\pm\scriptstyle 0.003$ | 0.823 $\pm\scriptstyle 0.002$   | 0.633 $\pm\scriptstyle 0.005$ |
>
> Q.3: What are experimental results if add a Normal Layer?
>
> A.3: As regards the addition of LayerNorm, experimental results on the effect of adding normalization layers were present in the original manuscript. We refer the reviewer to Appendix B.5 for our analysis. In Table 8, we show that the addition of LayerNorm layers in between the BiLSTM layers is detrimental on average, over all datasets.

---

> > ### Comment · Reviewer_mCBU · 2025-08-06
> >
> > Thank you for your detailed rebuttal. While I appreciate the additional experimentation you have done with GATs and the multilingual dataset, I maintain my original rate scores.
> >
> > My main concern remains the limited theoretical contribution. While this work demonstrates that normalization of biaffine score functions can reduce over-parameterization, this finding is limited. Its core finding, that normalization helps handle high-variance inputs to the softmax, aligns with well-established principles in deep learning.
> >
> > The expanded experiments strengthen the empirical validation, but they do not fundamentally change the nature of the contribution. The 85% parameter reduction is impressive, yet it primarily benefits a specific parsing architecture rather than providing broader insights applicable across different domains or tasks.
> >
> > While the work is technically solid, I believe the contribution falls short of the significance threshold typically expected for acceptance.

---

### Official Review · Reviewer_A3rQ · 2025-07-02

**Clarity:** 2
**Significance:** 3
**Originality:** 3
**Rating:** 4
**Confidence:** 3

**Summary:**

This paper investigates the impact of score normalization in biaffine dependency parsing models. This work argued that the current dependency parsers using biaffine scoring are over parameterized because they do NOT the score normalization which is typically used in Transformer in attention mechanisms. The paper provided theoretical justification through implicit regularization theory. It shows that deeper networks reduce weight matrix rank and consequently score variance. It demonstrated that the normalizing biaffine scores can achieve comparable or even better performance with up to 85% fewer parameters.

**Questions:**

No.

**Ethical Concerns:**

["NO or VERY MINOR ethics concerns only"]

**Limitations:**

The paper includes very limited discussions on limitations. I would suggest add some like the potential negative impact on datasets where current deep architectures are well-tuned, etc.

**Paper Formatting Concerns:**

No.

**Quality:**

3

**Strengths And Weaknesses:**

Strengths:
1. The connection between Transformer attention normalization and biaffine scoring is well-motivated.
2. The paper includes experiments on 6 diverse datasets, covering both semantic (ADE, CoNLL04, SciERC, ERFGC) and syntactic (enEWT, SciDTB) dependency parsing. It achieves similar performance with 85% parameter reduction. It includes systematic exploration on different layer depths, hyperparameters, architectural components, etc.

Weaknesses:
1. While Claim 1 provides intuition, the proof is somewhat informal. The connection between rank reduction and variance reduction could be more rigorously established.
2. The normalization benefits vary significantly across datasets. In addition, the main focus of the paper on BiLSTM-based architectures may limit applicability to modern Transformer-based parsers.

---

> ### Author Rebuttal · Authors · 2025-07-30
>
> Q.1: While Claim 1 provides intuition, the proof is somewhat informal. The connection between rank reduction and variance reduction could be more rigorously established.
>
> A.1: We thank the reviewer for pointing out that our proof (and claim) could be made more rigorous. We have restated the claim and proof formally with a more standard analysis of covariance using the trace. Please see below:
>
> *Claim 1* (Monotonic Increase of Output Variance with Rank)
>
> Let $X \in \mathbb{R}^n$ be a random vector with covariance matrix $\mathbf{K_{xx}} := \text{Cov}(X) \in \mathbb{R}^{n \times n}$, and let $\mathbf{A} \in \mathbb{R}^{m \times n}$ be a fixed matrix with singular value decomposition (SVD):
>
> $$\mathbf{A} = \sum_{i=1}^{min(m, n)} \sigma_i \mathbf{u}_i \mathbf{v}_t^{\top}$$
>
> where $\sigma_1 \geq \sigma_2 \geq ... \geq 0$. Define the best rank-$r$ approximation of $\mathbf{A}$ by truncated SVD as:
>
> $$\mathbf{A_r} := \sum_{i=1}^r \sigma_i \mathbf{u}_i \mathbf{v}_t^{\top}$$
>
> Let $Y_r = \mathbf{A}_rX \in \mathbb{R}^m$ denote the image of $X$ under the rank-$r$ linear map. Then, the total variance of $Y_r$, as measured by the trace of its covariance matrix, increases monotonically with $r$:
>
> $$\text{tr}(\text{Cov}(Y_r)) \leq \text{tr}(\text{Cov}(Y_{r + 1}))$$
>
> *Proof.* Let $\mathbf{K_{xx}} := \text{Cov}(X) \in \mathbb{R}^{n \times n}$ denote the covariance matrix of the random vector $X$. Since covariance matrices are symmetric and positive semi-definite (PSD), we have $\mathbf{K_{xx}} \succeq 0$. Let $\mathbf{A_r} := \sum_{i=1}^r \sigma_i \mathbf{u}_i \mathbf{v}_t^{\top}$ be the rank-$r$ truncated SVD of $\mathbf{A}$. Then the covariance matrix of $Y_r := \mathbf{A_r} X$ is:
>
> $$\text{Cov}(Y_r) = \mathbf{A_r} \mathbf{K_{xx}} \mathbf{A_r}^{\top}.$$
>
> To evaluate the total variance we compute the trace:
>
> $$\text{tr}\left(\text{Cov}(Y_r)\right) = \text{tr}\left(\mathbf{A_r}\mathbf{K_{xx}} \mathbf{A_r}^{\top}\right).$$
>
> Using the cyclic property of the trace $\left(\text{tr}(ABC) = \text{tr}(BCA)\right)$ and symmetry of $\mathbf{K_{xx}}$ we get:
>
> $$\text{tr}\left(\mathbf{A_r}\mathbf{K_{xx}} \mathbf{A_r}^{\top}\right) = \text{tr}\left(\mathbf{K_{xx}} \mathbf{A_r}^{\top} \mathbf{A_r} \right).$$
>
> Now define the matrix $\mathbf{M_r} := \mathbf{A_r}^{\top} \mathbf{A} \in \mathbb{R}^{n \times n}$, which is PSD. As $r$ increases, $\mathbf{A_r}$ includes more terms in its truncated SVD, so we have:
>
> $$\mathbf{M_{r}} = \sum_{i=1}^r \sigma^2_i \mathbf{v_i} \mathbf{v_i}^{\top}, \quad \mathbf{M_{r+1}} = \mathbf{M_r} + \sigma^2_{r+1} \mathbf{v_{r+1}} \mathbf{v_{r+1}}^{\top}.$$
>
> Thus:
>
> $$\mathbf{M}_{r+1} \succeq \mathbf{M}_r.$$
>
> Since $\mathbf{K_{xx}} \succeq 0$, and since the trace of a product of PSD matrices respects Loewner order (i.e., if $A \preceq B$, then $\text{tr}(CA) \leq \text{tr}(CB)$ for all $C \succeq 0$), we conclude:
>
> $$\text{tr}(\mathbf{K_{xx}} \mathbf{M_r}) \leq \text{tr}(\mathbf{K_{xx}}\mathbf{M_{r+1}}),$$
>
> which implies:
>
> $$\text{tr}\left(\text{Cov}(Y_r)\right) \leq \text{tr}\left(\text{Cov}(Y_{r+1})\right).$$
>
> Therefore, as the rank $r$ increases, the total variance $Y_r$ increases. $\square$
>
> Q.2.1: The normalization benefits vary significantly across datasets.
>
> A.2.1: Thank you for raising this point. In order to show the generalizability of our results, we have carried out six additional experiments on multilingual syntactic dependency parsing, showing  consistent results for a wide variety of languages.
>
> | score norm | $L_\psi$ | UD_Arabic-PADT    | UD_Chinese-GSD    | UD_Italian-ISDT   | UD_Japanese-GSD   | UD_Spanish-AnCora   | UD_Wolof-WTB      |
> |:---|:----|:------------------|:------------------|:------------------|:------------------|:--------------------|:------------------|
> | $\times$ | 0 | 0.538 $\pm\scriptstyle 0.005$ | 0.395 $\pm\scriptstyle 0.007$ | 0.563 $\pm\scriptstyle 0.006$ | 0.493 $\pm\scriptstyle 0.010$ | 0.554 $\pm\scriptstyle 0.004$   | 0.252 $\pm\scriptstyle 0.007$ |
> | $\times$ | 1 | 0.723 $\pm\scriptstyle 0.008$ | 0.653 $\pm\scriptstyle 0.007$ | 0.792 $\pm\scriptstyle 0.003$ | 0.812 $\pm\scriptstyle 0.003$ | 0.775 $\pm\scriptstyle 0.003$   | 0.525 $\pm\scriptstyle 0.006$ |
> | $\times$ | 2 | 0.745 $\pm\scriptstyle 0.004$ | 0.710 $\pm\scriptstyle 0.005$ | 0.826 $\pm\scriptstyle 0.003$ | 0.844 $\pm\scriptstyle 0.005$ | 0.807 $\pm\scriptstyle 0.002$   | 0.587 $\pm\scriptstyle 0.007$ |
> | $\times$ | 3 | 0.748 $\pm\scriptstyle 0.004$ | 0.717 $\pm\scriptstyle 0.007$ | 0.832 $\pm\scriptstyle 0.002$ | 0.849 $\pm\scriptstyle 0.003$ | 0.808 $\pm\scriptstyle 0.002$   | 0.614 $\pm\scriptstyle 0.005$ |
> | $\checkmark$ | 0 | 0.609 $\pm\scriptstyle 0.003$ | 0.479 $\pm\scriptstyle 0.007$ | 0.633 $\pm\scriptstyle 0.002$ | 0.585 $\pm\scriptstyle 0.005$ | 0.620 $\pm\scriptstyle 0.002$   | 0.305 $\pm\scriptstyle 0.005$ |
> | $\checkmark$ | 1 | 0.737 $\pm\scriptstyle 0.007$ | 0.693 $\pm\scriptstyle 0.003$ | 0.820 $\pm\scriptstyle 0.004$ | 0.838 $\pm\scriptstyle 0.004$ | 0.801 $\pm\scriptstyle 0.002$   | 0.556 $\pm\scriptstyle 0.003$ |
> | $\checkmark$ | 2 | 0.758 $\pm\scriptstyle 0.003$ | 0.736 $\pm\scriptstyle 0.005$ | 0.842 $\pm\scriptstyle 0.004$ | 0.859 $\pm\scriptstyle 0.004$ | 0.822 $\pm\scriptstyle 0.001$   | 0.613 $\pm\scriptstyle 0.008$ |
> | $\checkmark$ | 3 | 0.759 $\pm\scriptstyle 0.005$ | 0.742 $\pm\scriptstyle 0.006$ | 0.845 $\pm\scriptstyle 0.002$ | 0.859 $\pm\scriptstyle 0.003$ | 0.823 $\pm\scriptstyle 0.002$   | 0.633 $\pm\scriptstyle 0.005$ |
>
> Furthermore, we believe it is important to keep into account that we found the difference in performance to be statistically significant on all datasets, as mentioned in Figure 3.
>
> Q.2.2: In addition, the main focus of the paper on BiLSTM-based architectures may limit applicability to modern Transformer-based parsers.
>
> A.2.2: We would like to point out that the parser with which we experiment is Transformer-based, with the Dozat and Manning biaffine layer still being the standard for dependency parsing. Even without using BiLSTM layers and fully fine-tuning the Transformer, we have found that normalization provides statistically significant benefits, as shown in Figure 4 for the SciERC dataset. In Appendix B.6, we also show this more generally on all four SemDP datasets with BERT-base, BERT-large, DeBERTa-base, and DeBERTa-large. The effect is particularly pronounced for DeBERTa-base.

---

### Official Review · Reviewer_MP7r · 2025-07-03

**Clarity:** 3
**Significance:** 2
**Originality:** 3
**Rating:** 3
**Confidence:** 5

**Summary:**

This work presents theoretical evidence and empirical results for the influence of scaling the scores produced by biaffine transformations in dependency parsing tasks. The authors find that the score variance produced by a lack of score scaling hurts model performance
when predicting edges and relations. Experimental results demonstrate that dependency parsing models equipped with normalization can obtain better performance with substantially fewer trained parameters.

**Questions:**

See the weaknesses

**Ethical Concerns:**

["NO or VERY MINOR ethics concerns only"]

**Final Justification:**

The work lags behind the current state of the art in dependency parsing.

**Limitations:**

yes

**Quality:**

2

**Strengths And Weaknesses:**

Strengths:
1. Biaffine score function is a common and default component in dependency parsing tasks. The authors are very careful and aware of the potential problem in it. They analyze the relation between normalization and parameter quantity from theoretical and experimental perspectives.
2. This paper conduct extensive experiments to verify their claim.

Weaknesses:
1. I wonder whether this finding is still effective in other languages, especially low-resource languages.
2. This research question is too minor to have a significant impact on the broad research community. I think the main research content of this paper is not suitable for publication in NeurIPS. I recommend that the author pay attention to more targeted venues, so that researchers in specific sub-community can pay attention to this work.
3. Please change eh, ed, rh, and rd in figure 2 to the format of mathematical vector (i.e., $e^h$, $e^d$, $r^h$ and $r^d$), so they are consistent with the text and equation.

---

> ### Author Rebuttal · Authors · 2025-07-30
>
> Q.1: I wonder whether this finding is still effective in other languages, especially low-resource languages.
>
> A.1: We thank the reviewer for suggesting multilingual settings. We have run the model on six non-English UD datasets (UD_Arabic-PADT, UD_Chinese-GSD, UD_Italian-ISDT, UD_Japanese-GSD, UD_Spanish-AnCora, UD_Wolof-WTB) and verified that our findings are consistent. In the following we summarized the multilingual results. We have added them to the paper.
>
> | score norm | $L_\psi$ | UD_Arabic-PADT    | UD_Chinese-GSD    | UD_Italian-ISDT   | UD_Japanese-GSD   | UD_Spanish-AnCora   | UD_Wolof-WTB      |
> |:---|:----|:------------------|:------------------|:------------------|:------------------|:--------------------|:------------------|
> | $\times$ | 0 | 0.538 $\pm\scriptstyle 0.005$ | 0.395 $\pm\scriptstyle 0.007$ | 0.563 $\pm\scriptstyle 0.006$ | 0.493 $\pm\scriptstyle 0.010$ | 0.554 $\pm\scriptstyle 0.004$   | 0.252 $\pm\scriptstyle 0.007$ |
> | $\times$ | 1 | 0.723 $\pm\scriptstyle 0.008$ | 0.653 $\pm\scriptstyle 0.007$ | 0.792 $\pm\scriptstyle 0.003$ | 0.812 $\pm\scriptstyle 0.003$ | 0.775 $\pm\scriptstyle 0.003$   | 0.525 $\pm\scriptstyle 0.006$ |
> | $\times$ | 2 | 0.745 $\pm\scriptstyle 0.004$ | 0.710 $\pm\scriptstyle 0.005$ | 0.826 $\pm\scriptstyle 0.003$ | 0.844 $\pm\scriptstyle 0.005$ | 0.807 $\pm\scriptstyle 0.002$   | 0.587 $\pm\scriptstyle 0.007$ |
> | $\times$ | 3 | 0.748 $\pm\scriptstyle 0.004$ | 0.717 $\pm\scriptstyle 0.007$ | 0.832 $\pm\scriptstyle 0.002$ | 0.849 $\pm\scriptstyle 0.003$ | 0.808 $\pm\scriptstyle 0.002$   | 0.614 $\pm\scriptstyle 0.005$ |
> | $\checkmark$ | 0 | 0.609 $\pm\scriptstyle 0.003$ | 0.479 $\pm\scriptstyle 0.007$ | 0.633 $\pm\scriptstyle 0.002$ | 0.585 $\pm\scriptstyle 0.005$ | 0.620 $\pm\scriptstyle 0.002$   | 0.305 $\pm\scriptstyle 0.005$ |
> | $\checkmark$ | 1 | 0.737 $\pm\scriptstyle 0.007$ | 0.693 $\pm\scriptstyle 0.003$ | 0.820 $\pm\scriptstyle 0.004$ | 0.838 $\pm\scriptstyle 0.004$ | 0.801 $\pm\scriptstyle 0.002$   | 0.556 $\pm\scriptstyle 0.003$ |
> | $\checkmark$ | 2 | 0.758 $\pm\scriptstyle 0.003$ | 0.736 $\pm\scriptstyle 0.005$ | 0.842 $\pm\scriptstyle 0.004$ | 0.859 $\pm\scriptstyle 0.004$ | 0.822 $\pm\scriptstyle 0.001$   | 0.613 $\pm\scriptstyle 0.008$ |
> | $\checkmark$ | 3 | 0.759 $\pm\scriptstyle 0.005$ | 0.742 $\pm\scriptstyle 0.006$ | 0.845 $\pm\scriptstyle 0.002$ | 0.859 $\pm\scriptstyle 0.003$ | 0.823 $\pm\scriptstyle 0.002$   | 0.633 $\pm\scriptstyle 0.005$ |
>
> Q.2: This research question is too minor to have a significant impact on the broad research community. I think the main research content of this paper is not suitable for publication in NeurIPS. I recommend that the author pay attention to more targeted venues, so that researchers in specific sub-community can pay attention to this work.
>
> A.2: While it is true that semantic and syntactic dependency parsing are specific to communities in NLP, our paper contributes towards analyzing normalization phenomena in architectures used broadly in graph inference tasks (modeling edges as fully connected weighted graphs via attention mechanisms such as biaffine transformations, see [1,2,3] for reference). To further demonstrate this, we have conducted an experiment using Graph Attention Networks (GAT) by adding a stack of biaffine/GAT pairs. The purpose of this is to encode higher-order dependencies before the final biaffine layer. The results with this new architecture are consistent, as shown in the table below.
>
> |   norm |   $L_\phi$ | CoNLL04           | ADE               | SciERC            | enEWT             | SciDTB             | ERFGC             |
> |-------:|----:|:------------------|:------------------|:------------------|:------------------|:------------------|:------------------|
> |      $\times$ |   0 | 0.526 $\pm 0.046$ | 0.517 $\pm 0.038$ | 0.123 $\pm 0.050$ | 0.610 $\pm 0.009$ | 0.727 $\pm 0.006$ | 0.536 $\pm 0.013$ |
> |      $\times$ |   1 | 0.493 $\pm 0.037$ | 0.509 $\pm 0.022$ | 0.039 $\pm 0.020$ | 0.583 $\pm 0.011$ | 0.731 $\pm 0.009$ | 0.599 $\pm 0.008$ |
> |      $\times$ |   2 | 0.485 $\pm 0.033$ | 0.509 $\pm 0.025$ | 0.029 $\pm 0.041$ | 0.540 $\pm 0.012$ | 0.710 $\pm 0.013$ | 0.611 $\pm 0.015$ |
> |      $\times$ |   3 | 0.481 $\pm 0.037$ | 0.487 $\pm 0.040$ | 0.000 $\pm 0.000$ | 0.511 $\pm 0.015$ | 0.700 $\pm 0.010$ | 0.585 $\pm 0.009$ |
> |      $\checkmark$ |   0 | 0.574 $\pm 0.028$ | 0.556 $\pm 0.037$ | 0.156 $\pm 0.031$ | 0.671 $\pm 0.007$ | 0.778 $\pm 0.006$ | 0.607 $\pm 0.009$ |
> |      $\checkmark$ |   1 | 0.550 $\pm 0.036$ | 0.587 $\pm 0.015$ | 0.148 $\pm 0.028$ | 0.696 $\pm 0.005$ | 0.809 $\pm 0.008$ | 0.639 $\pm 0.007$ |
> |      $\checkmark$ |   2 | 0.563 $\pm 0.029$ | 0.534 $\pm 0.023$ | 0.128 $\pm 0.029$ | 0.657 $\pm 0.007$ | 0.795 $\pm 0.008$ | 0.637 $\pm 0.007$ |
> |      $\checkmark$ |   3 | 0.514 $\pm 0.026$ | 0.543 $\pm 0.024$ | 0.071 $\pm 0.021$ | 0.596 $\pm 0.010$ | 0.770 $\pm 0.007$ | 0.616 $\pm 0.006$ |
>
> Similar to BiLSTMs, the normalization improves the scores in the case of GATs as well, showing that our method generalizes to other architectures, besides Transformers and BiLSTMS. We have included these results to the paper.
>
> In addition to our technical contribution, we believe that our work contributes to the broad NeurIPS community, by suggesting the importance of peculiar settings such as normalization in the training of neural network models. Moreover, we would like to respectfully point out that the NeurIPS guideline of acceptance suggests a work with "high impact on at least one sub-area of AI" (per grade 5), meaning that broad impact is not strictly required for acceptance. We appreciate the reviewer's consideration in this regard.
>
> - [1] Kazi et al. "Differentiable Graph Module (DGM) for Graph Convolutional Networks". 2023.
>
> - [2] P. Velickovic, G. Cucurull, A. Casanova, A. Romero, P. Lio, and Y. Bengio, "Graph attention networks". 2017.
>
> - [3] J. Zhang, X. Shi, J. Xie, H. Ma, I. King, and D.-Y. Yeung, "GAAN: Gated attention networks for learning on large and spatiotemporal graphs". 2018.
>
> Q.3:  Please change eh, ed, rh, and rd in figure 2 to the format of mathematical vector (i.e., $e^h$, $e^d$, $r^h$, and $r^d$), so they are consistent with the text and equation.
>
> A.3: We have amended the formatting oversight as requested.

---

### Official Review · Reviewer_BR5E · 2025-07-05

**Clarity:** 4
**Significance:** 3
**Originality:** 3
**Rating:** 5
**Confidence:** 4

**Summary:**

It seems that normalizing biaffine scores for BiLSTMs gives significant improvement in terms of performance and parameter efficiency. This claim is supported by running dependency parsing task on multiple datasets, with several ablation studies. A possible reason for this is also discussed in section 3.

**Questions:**

- We see in some cases, e.g. in Figure 5 that the normalized version seems to drop in performance compared to the raw version as number of layers is increased. Do you have any guesses of why that might be?

**Ethical Concerns:**

["NO or VERY MINOR ethics concerns only"]

**Final Justification:**

I thank authors for additional information provided.

I have read other reviews and I see the main concerns, and here are my thoughts about them:

- Results on other languages e.g. low resource ones

Although authors provide some results on more languages, I believe that is a separate question. The paper is good enough contribution only for high-resource languages

- The contribution being simple

Yes perhaps, it is quite simple. I believe that is not grounds for rejection. And simple solutions should be published to benefit the wider ML / NLP community.

- The contribution being too niche and perhaps better suited for more niche venues

Maybe. I am not aware if it's kind of outside the area of NeurIPS.

**Limitations:**

We don't see a separate section for limitations, but in my opinion limitations are adequately and honestly discussed. Some with other text (e.g. size of dataset and kind of dataset when introduced), some in conclusions (last paragraph), and some in appendix.

**Paper Formatting Concerns:**

- see line 38. you probably meant to write Std([something])?
- line 208 it says "UD". The words UD and enEWT are sometimes interchangeably used. The table referred uses enEWT.
- it might be better for some graphs if possible to use false origin on y-axis to see better e.g. in figure 7. This is only a suggestion if easily possible.

**Quality:**

3

**Strengths And Weaknesses:**

**Strengths**
- one possible cause of this result discussed sufficiently. Claim: maybe more layers cause variance to be in check and normalization is not much needed as small singular values decay.
- many datasets utilized that have dependency parsing of different kinds (recipes, news, drug adverse effects, etc.)
- normalizing shown to help with thorough experiments, with significance of outcomes measures satisfactorily
- parameter efficiency for one particular case shown to be up to 85% more with normalization where with normalization only one layer is enough to match the performance of three without normalization


**Weaknesses**
- for now only seen for dependency parsing task
- while a possible reason for raw version performing okay with higher number of layers is discussed, it is not known why in some cases the performance of normalized version drops and exhibits high variance when number of layers is increased (e.g. Appendix, line 489) [See Question related to this as well]
- the gain for tagging tasks not that significant, but this adequately discussed

---

> ### Author Rebuttal · Authors · 2025-07-30
>
> We thank the reviewer for their insightful review.
>
> Q.1: for now only seen for dependency parsing task
>
> A.1: We acknowledge that our work targets semantic and syntactic dependency parsing tasks only. This was purposeful because this variant (Dozat and Manning) of biaffine scoring is widespread on these tasks in particular. While we plan to extend our normalization analysis to graph inference broadly (e.g. using QM9 for small molecules or citation networks), we believe that the current work presents a solid and focused contribution using one of the most common and SOTA architectures. Moreover, extending our analysis to non-linguistic tasks would involve many architectural variants (e.g. GNN types) that we feel are necessary to cover in a separate work built on the contribution of the current work. As an initial step towards using Graph Attention Networks (GAT), we have repeated the experiment by adding a stack of biaffine/GAT pairs to encode higher-order dependencies before the final biaffine layer. We achieve consistent results as shown in the table below. The performance increases for $L_\phi \in \{1, 2\}$, compared to $L_\phi = 0$, but drops again at $L_\phi = 3$:
>
> |   norm |   $L_\phi$ | CoNLL04           | ADE               | SciERC            | enEWT             | SciDTB             | ERFGC             |
> |-------:|----:|:------------------|:------------------|:------------------|:------------------|:------------------|:------------------|
> |      n |   0 | 0.526 $\pm 0.046$ | 0.517 $\pm 0.038$ | 0.123 $\pm 0.050$ | 0.610 $\pm 0.009$ | 0.727 $\pm 0.006$ | 0.536 $\pm 0.013$ |
> |      n |   1 | 0.493 $\pm 0.037$ | 0.509 $\pm 0.022$ | 0.039 $\pm 0.020$ | 0.583 $\pm 0.011$ | 0.731 $\pm 0.009$ | 0.599 $\pm 0.008$ |
> |      n |   2 | 0.485 $\pm 0.033$ | 0.509 $\pm 0.025$ | 0.029 $\pm 0.041$ | 0.540 $\pm 0.012$ | 0.710 $\pm 0.013$ | 0.611 $\pm 0.015$ |
> |      n |   3 | 0.481 $\pm 0.037$ | 0.487 $\pm 0.040$ | 0.000 $\pm 0.000$ | 0.511 $\pm 0.015$ | 0.700 $\pm 0.010$ | 0.585 $\pm 0.009$ |
> |      y |   0 | 0.574 $\pm 0.028$ | 0.556 $\pm 0.037$ | 0.156 $\pm 0.031$ | 0.671 $\pm 0.007$ | 0.778 $\pm 0.006$ | 0.607 $\pm 0.009$ |
> |      y |   1 | 0.550 $\pm 0.036$ | 0.587 $\pm 0.015$ | 0.148 $\pm 0.028$ | 0.696 $\pm 0.005$ | 0.809 $\pm 0.008$ | 0.639 $\pm 0.007$ |
> |      y |   2 | 0.563 $\pm 0.029$ | 0.534 $\pm 0.023$ | 0.128 $\pm 0.029$ | 0.657 $\pm 0.007$ | 0.795 $\pm 0.008$ | 0.637 $\pm 0.007$ |
> |      y |   3 | 0.514 $\pm 0.026$ | 0.543 $\pm 0.024$ | 0.071 $\pm 0.021$ | 0.596 $\pm 0.010$ | 0.770 $\pm 0.007$ | 0.616 $\pm 0.006$ |
>
> **The normalization improves the scores in the case of GATs as well, showing that our method generalizes to other architectures, besides Transformers and BiLSTMS. We have included these results to the paper.**
>
> Q.2: while a possible reason for raw version performing okay with higher number of layers is discussed, it is not known why in some cases the performance of normalized version drops and exhibits high variance when number of layers is increased
>
> A.2: We have repeated the experiment with $L_\psi > 3$ over three seeds and increased the amount of training steps to 20,000. All of the models made it to at least 6k steps (but not all of them made it to 8k or further, due to early stopping). We show the results at 2k, 4k, and 6k steps below (**rows with no high variances removed due to character limit**):
>
> 2000 steps:
>
> | norm   | $L_\psi$   | CoNLL04           | ADE               | SciERC            | enEWT             | SciDTB             | ERFGC             |
> |---:|---:|:------------------|:------------------|:------------------|:------------------|:------------------|:------------------|
> |  n |  7 | 0.606 $\pm 0.012$ | 0.731 $\pm 0.004$ | 0.260 $\pm 0.026$ | 0.811 $\pm 0.007$ | 0.898 $\pm 0.006$ | 0.698 $\pm 0.006$ |
> |  n |  8 | 0.594 $\pm 0.012$ | 0.712 $\pm 0.012$ | 0.248 $\pm 0.002$ | 0.783 $\pm 0.003$ | 0.882 $\pm 0.007$ | 0.692 $\pm 0.010$ |
> |  n | 10 | 0.565 $\pm 0.012$ | 0.701 $\pm 0.018$ | 0.233 $\pm 0.022$ | 0.718 $\pm 0.049$ | 0.839 $\pm 0.034$ | 0.441 $\pm 0.312$ |
> |  y |  9 | 0.215 $\pm 0.289$ | 0.229 $\pm 0.324$ | 0.260 $\pm 0.039$ | 0.791 $\pm 0.006$ | 0.883 $\pm 0.007$ | 0.450 $\pm 0.311$ |
> |  y | 10 | 0.386 $\pm 0.261$ | 0.622 $\pm 0.018$ | 0.168 $\pm 0.120$ | 0.769 $\pm 0.016$ | 0.879 $\pm 0.009$ | 0.648 $\pm 0.006$ |
>
> 4000 steps:
>
> | norm   | $L_\psi$   | CoNLL04           | ADE               | SciERC            | enEWT             | SciDTB             | ERFGC             |
> |---:|---:|:------------------|:------------------|:------------------|:------------------|:------------------|:------------------|
> |  n |  7 | 0.551 $\pm 0.150$ | 0.719 $\pm 0.017$ | 0.284 $\pm 0.032$ | 0.833 $\pm 0.022$ | 0.907 $\pm 0.010$ | 0.700 $\pm 0.005$ |
> |  n |  8 | 0.507 $\pm 0.227$ | 0.711 $\pm 0.008$ | 0.271 $\pm 0.023$ | 0.813 $\pm 0.030$ | 0.895 $\pm 0.014$ | 0.694 $\pm 0.008$ |
> |  n | 10 | 0.588 $\pm 0.036$ | 0.701 $\pm 0.033$ | 0.262 $\pm 0.035$ | 0.755 $\pm 0.059$ | 0.865 $\pm 0.037$ | 0.442 $\pm 0.313$ |
> |  y |  9 | 0.279 $\pm 0.270$ | 0.244 $\pm 0.322$ | 0.289 $\pm 0.041$ | 0.813 $\pm 0.022$ | 0.894 $\pm 0.012$ | 0.457 $\pm 0.311$ |
> |  y | 10 | 0.399 $\pm 0.278$ | 0.651 $\pm 0.032$ | 0.165 $\pm 0.123$ | 0.797 $\pm 0.030$ | 0.890 $\pm 0.014$ | 0.656 $\pm 0.009$ |
>
> 6000 steps:
>
> | norm   | $L_\psi$   | CoNLL04           | ADE               | SciERC            | enEWT             | SciDTB             | ERFGC             |
> |---:|---:|:------------------|:------------------|:------------------|:------------------|:------------------|:------------------|
> |  n |  7 | 0.581 $\pm 0.130$ | 0.712 $\pm 0.021$ | 0.307 $\pm 0.041$ | 0.846 $\pm 0.026$ | 0.910 $\pm 0.010$ | 0.702 $\pm 0.006$ |
> |  n |  8 | 0.466 $\pm 0.254$ | 0.708 $\pm 0.009$ | 0.292 $\pm 0.039$ | 0.829 $\pm 0.033$ | 0.901 $\pm 0.015$ | 0.695 $\pm 0.011$ |
> |  n | 10 | 0.592 $\pm 0.036$ | 0.549 $\pm 0.295$ | 0.288 $\pm 0.047$ | 0.779 $\pm 0.062$ | 0.877 $\pm 0.035$ | 0.448 $\pm 0.317$ |
> |  y |  9 | 0.329 $\pm 0.282$ | 0.242 $\pm 0.328$ | 0.310 $\pm 0.047$ | 0.827 $\pm 0.027$ | 0.899 $\pm 0.012$ | 0.455 $\pm 0.311$ |
> |  y | 10 | 0.410 $\pm 0.280$ | 0.669 $\pm 0.037$ | 0.187 $\pm 0.141$ | 0.813 $\pm 0.034$ | 0.896 $\pm 0.014$ | 0.658 $\pm 0.010$ |
>
> The results show that once a model fails to find a good trajectory at the start, it likely remains on a poor trajectory. For example, the model with $L_\psi \in \{7, 8\}$ and norm = n does fine on CoNNL04 at 2,000 steps; however, one of the seeds then fails randomly at 4,000 steps and the performance remains poor past that point. For ADE, we can see that model performance with $L_\psi = 10 $ and norm = n is fine at 2k and 4k steps, but suddenly drops with higher variance at 6k steps. Once again, this is because the model suddenly breaks and then remains broken for a specific seed. The table below does not show some of the poor performances because it reports the results of each seed at the best validation performance, up to 20k steps:
>
> | norm   | $L_\psi$   | CoNLL04           | ADE               | SciERC            | enEWT             | SciDTB             | ERFGC             |
> |---:|---:|:------------------|:------------------|:------------------|:------------------|:------------------|:------------------|
> |  n |  7 | 0.668 $\pm 0.015$ | 0.722 $\pm 0.016$ | 0.380 $\pm 0.032$ | 0.894 $\pm 0.002$ | 0.929 $\pm 0.002$ | 0.706 $\pm 0.003$ |
> |  n |  8 | 0.625 $\pm 0.028$ | 0.707 $\pm 0.025$ | 0.392 $\pm 0.016$ | 0.893 $\pm 0.002$ | 0.926 $\pm 0.002$ | 0.699 $\pm 0.007$ |
> |  n | 10 | 0.635 $\pm 0.038$ | 0.720 $\pm 0.011$ | 0.373 $\pm 0.016$ | 0.879 $\pm 0.007$ | 0.917 $\pm 0.002$ | 0.454 $\pm 0.321$ |
> |  y |  9 | 0.454 $\pm 0.274$ | 0.511 $\pm 0.313$ | 0.384 $\pm 0.024$ | 0.880 $\pm 0.001$ | 0.916 $\pm 0.001$ | 0.457 $\pm 0.306$ |
> |  y | 10 | 0.478 $\pm 0.283$ | 0.695 $\pm 0.022$ | 0.237 $\pm 0.169$ | 0.881 $\pm 0.001$ | 0.911 $\pm 0.002$ | 0.662 $\pm 0.015$ |
>
> Despite the random crashes, increasing the number of steps increased the performance and reduced the Std for deeper stacks, compared to the original manuscript. While we still see instability for CoNLL04 and SciERC, it is reduced compared to the results shown in the original manuscript in Figure 5. Furthermore, the high variance almost disappears for ADE and ERFGC when training for a higher number of steps. Even more importantly, the results are identical for enEWT and SciDTB, with and without normalization, hinting at the fact that this is not inherently a problem of the model, but has to do with the nature of the datasets.
> More specifically, we also see some instability for ERFGC **without** score normalization at $L_\psi = 10$, and at $L_\psi = 9$ with score normalization. Similarly high variance can be seen for ADE with normalization at $L_\psi = 9$ (but not 10). Given that the high variances are produced by a few zero or near-zero scores, the instability seems to mostly be a product of the higher number of layers.
>
> We have also analyzed the behavior of the gradients w.r.t. the depth of the layers. For SciERC (highest Std at $L_\psi = 10$ in the original manuscript), the table below shows the head and tail for 14k steps of norm of the gradient for the 10 layers:
>
> |  step   |     0 |     1 |     2 |     3 |     4 |     5 |     6 |     7 |     8 |     9 |
> |------:|------:|------:|------:|------:|------:|------:|------:|------:|------:|------:|
> |     0 | 0.002 | 0.002 | 0.004 | 0.008 | 0.020 | 0.048 | 0.115 | 0.279 | 0.638 | 1.607 |
> |  ...  |       |       |       |       |       |       |       |       |       |       |
> | 13999 | 0.070 | 0.039 | 0.018 | 0.014 | 0.013 | 0.010 | 0.012 | 0.013 | 0.036 | 0.072 |
>
> Since the gradient evens out, we reckon it not to be the issue at hand here. It is likely that the higher parameter count of 10 layers simply necessitates more training steps.
>
> Formatting:
>
> We commit to fix the issues raised by the reviewer w.r.t. line 38, line 208, and commit to improving the readability of Figs. 6 and 7.

---

### Note · Authors · 2025-08-15

We wish to thank the reviewers for their observations and suggestions.

We respectfully disagree with the comment that our results are not of interest to a broader community. Link prediction is a fundamental task in graph machine learning and the use of biaffine layers is common practice. While our focus was on dependency parsing in NLP to substantiate our claims, our results hold for other domains as well.

We have carried out experiments to show that normalization helps link prediction tasks on three non-linguistic datasets: PCQM-Contact [1], CIFAR10 Superpixel [2], and QM9 [3]. PCQM-Contact and QM9 are molecule datasets, while CIFAR10 Superpixel is a dataset of superpixels clusters constructed from CIFAR10.

We feed graph node features into a GAT with $L_\phi \in \{1, 2, 3\}$ layers and provide it with a fully-connected adjacency matrix without any edge attributes. The processed node features are passed into the biaffine layer to produce edge predictions. The unlabeled F1 scores are reported in the table below.


|   Norm     | $L_\phi$ | PCQM-Contact             | CIFAR10 Superpixel       | QM9                      |
|-----------:|-----------------:|:-------------------------|:-------------------------|:-------------------------|
|          n |                1 | 0.241 $\pm 0.044$ | 0.407 $\pm 0.110$ | 0.861 $\pm 0.008$ |
|          n |                2 | 0.217 $\pm 0.019$ | 0.528 $\pm 0.134$ | 0.876 $\pm 0.006$ |
|          n |                3 | 0.245 $\pm 0.034$ | 0.751 $\pm 0.014$ | 0.877 $\pm 0.008$ |
|          y |                1 | 0.288 $\pm 0.090$ | 0.531 $\pm 0.038$ | 0.886 $\pm 0.001$ |
|          y |                2 | 0.209 $\pm 0.028$ | 0.585 $\pm 0.121$ | 0.880 $\pm 0.004$ |
|          y |                3 | 0.237 $\pm 0.019$ | 0.703 $\pm 0.046$ | 0.881 $\pm 0.013$ |

In all cases, we can see that the gap in performance is large at $L_\phi = 1$ and decreases (or flips) with additional layers. This is an especially important result considering it holds even with such a NN consisting of just 100-160k parameters.

[1] Dwivedi et al. (2022). Long range graph benchmark. Advances in Neural Information Processing Systems

[2] Dwivedi et al. (2023). Benchmarking graph neural networks. Journal of Machine Learning Research

[3] Ramakrishnan et al. (2014). Quantum chemistry structures and properties of 134 kilo molecules. Scientific data

---

### Decision · Program_Chairs · 2025-09-17

**Decision:**

Accept (poster)

**Comment:**

The paper presents results of link prediction, in particular for dependency parsing, after implementation of score normalisation.  The authors motivate the normalisation theoretically based on architecture and exchange out a corresponding module and its required parameters, arguing that lack of normalisation resulted in overparamterisation.  In the rebuttal, the authors also showed that their trade-off is also valid for other (non-linguistic) tasks.  While the paper is heavily motivated with the dependency parsing task, it is clear that the observations are about network topology and not limited to this task.  While the paper aligns with well-established principles in deep learning, the reviewers do not point out any redundancy in this particular finding.  The task itself, however, dependency parsing, is a central task to NLP.  For researchers without access to compute resources beyond a laptop, an 85% reduction in parameters is of central importance.